# Generalizable Person Re-identification via Balancing Alignment and Uniformity

**Yoonki Cho**   **Jaeyoon Kim**   **Woo Jae Kim**   **Junsik Jung**   **Sung-Eui Yoon**

KAIST

## Abstract

Domain generalizable person re-identification (DG re-ID) aims to learn discriminative representations that are robust to distributional shifts. While data augmentation is a straightforward solution to improve generalization, certain augmentations exhibit a polarized effect in this task, enhancing in-distribution performance while deteriorating out-of-distribution performance. In this paper, we investigate this phenomenon and reveal that it leads to sparse representation spaces with reduced uniformity. To address this issue, we propose a novel framework, Balancing Alignment and Uniformity (BAU), which effectively mitigates this effect by maintaining a balance between alignment and uniformity. Specifically, BAU incorporates alignment and uniformity losses applied to both original and augmented images and integrates a weighting strategy to assess the reliability of augmented samples, further improving the alignment loss. Additionally, we introduce a domain-specific uniformity loss that promotes uniformity within each source domain, thereby enhancing the learning of domain-invariant features. Extensive experimental results demonstrate that BAU effectively exploits the advantages of data augmentation, which previous studies could not fully utilize, and achieves state-of-the-art performance without requiring complex training procedures. The code is available at https://github.com/yoonkicho/BAU.

## 1   Introduction

Person re-identification (re-ID) aims to match a person with the same identity as a given query across disjoint camera views and different timestamps [88, 96]. Thanks to the discriminative features learned from deep neural networks, significant achievements have been made in this task [1, 44, 45, 72, 73, 78]. However, these learned feature spaces rely on the assumption of independent and identically distributed (*i.i.d.*) training and testing data, which leads to substantial performance degradation in unseen domains with distributional shifts. To address this issue, domain generalizable person re-ID (DG re-ID) has emerged, focusing on learning representations that are robust to domain shifts [71].

Existing DG re-ID methods often leverage advanced network architectures, such as feature normalization modules [34–36, 53, 107], domain-specific designs [12, 86, 90], and the integration of transformers [51, 60]. Alternatively, some approaches employ domain adversarial training [6, 20, 91] or meta-learning strategies [10, 40, 71, 94] to learn domain-invariant representations across source domains. Although these studies have shown promising results, they often involve complex training procedures that require significant engineering effort or are prone to training instability [3, 62, 66].

On the other hand, data augmentation is a straightforward solution to enhance generalization capability by simulating diverse data variations during training. Due to its simplicity and effectiveness, numerous efforts have been made to adopt this approach for various DG tasks [69, 75, 93, 99, 104, 105]. However, in the context of DG re-ID, some data augmentations have been observed to exhibit a polarized effect – improving performance in the source domain while potentially degrading it in the

38th Conference on Neural Information Processing Systems (NeurIPS 2024).

target domain. A notable example is Random Erasing [101], a technique widely used in person re-ID, which has been shown to deteriorate cross-domain re-ID performance [34, 56, 94]. Despite this observation, the underlying causes and potential solutions for this phenomenon remain underexplored.

In this paper, we first investigate the polarized effect of data augmentations in DG re-ID. Recent studies have shown that alignment and uniformity in the representation space are closely related to feature generalizability [21, 49, 57, 65, 80, 81]. Building upon this, we reveal that data augmentations can induce sparse representation spaces with less uniformity, which may be detrimental to the open-set nature of the re-ID task, where learning diverse visual information is crucial for generalization [7, 12, 58, 87]. Based on our analysis, we propose a simple yet effective framework, Balancing Alignment and Uniformity (BAU), which alleviates the polarized effect of data augmentations by maintaining a balance between alignment and uniformity. Specifically, it regularizes the representation space by applying alignment and uniformity losses to both original and augmented images. Additionally, we introduce a weighting strategy that considers the reliability of augmented samples to improve the alignment loss. We further propose a domain-specific uniformity loss to promote uniformity within each source domain, enhancing the learning of domain-invariant features. Consequently, BAU effectively exploits the advantages of data augmentation, which previous studies could not fully utilize, and achieves state-of-the-art performance on various benchmarks.

In summary, our contributions are as follows:

- We investigate the polarized effect of data augmentations in DG re-ID and reveal that they can lead to sparse representation spaces, which are detrimental to generalization.
- We propose a novel BAU framework that mitigates the polarized effect of data augmentations by balancing alignment and uniformity in the representation space. Additionally, we introduce a domain-specific uniformity loss to enhance the learning of domain-invariant representations.
- Through extensive experiments on various benchmarks and protocols, we demonstrate that BAU achieves state-of-the-art performance, even without complex training procedures.

## 2 Related Work

**Generalizable Person Re-identification.** Domain generalizable person re-identification (DG Re-ID) focuses on learning discriminative representations for person retrieval that are robust across unseen domains with distributional shifts. Significant efforts have been made in this task [2, 4, 17, 52, 60, 71, 82] due to its practicality, as it does not require additional model updates for target data, unlike domain adaptation approaches [19, 22, 23, 54, 102]. Given that learned feature statistics can be biased toward the source domain [5, 24, 47, 76], various feature normalization methods [10, 27, 34, 35, 107] have been proposed to mitigate this domain bias. For instance, SNR [36] eliminates style bias through feature disentanglement, and GDNorm [53] refines feature statistics using a Gaussian process. To achieve domain-invariant representations, several studies [10, 61, 71, 90] leverage domain-adversarial training [20, 41] or meta-learning [18, 40]. DDAN [6] utilizes domain-wise adversarial feature learning to reduce domain discrepancies for domain invariance. $M^3L$ [94] employs meta-learning to simulate the train-test process of domain generalization with multi-source datasets. There have also been attempts [50, 51] that explore advanced matching strategies between query and gallery images for retrieval to improve interpretability and generalization performance. Recently, Mixture-of-Experts (MoE) based approaches [12, 33] have emerged, where multiple domain-specific experts are trained and applied to the target domain, with META [86] alleviating the model scalability issue by domain-specific batch normalization [5]. In contrast, we effectively utilize the diversity provided by data augmentations to enhance generalization without relying on advanced network architectures or encountering training instability associated with adversarial or meta-learning [3, 62, 66].

**Alignment and Uniformity.** Wang and Isola [81] proposed that contrastive learning encompasses two main objectives: alignment, which aims to learn similar representations for positive pairs, and uniformity, which strives to distribute representations uniformly on the unit hypersphere. This framework has significantly influenced representation learning by potentially indicating the feature generalizability [49, 57, 63, 65, 77, 80]. For instance, the concepts of alignment and uniformity have been extensively studied to learn robust representations for improved generalizability across various downstream tasks [21, 57, 77] or domains [65, 80]. This approach has also proven effective when applied to multiple data modalities, including images and text [49, 63]. Despite these advances, the potential of alignment and uniformity in addressing the challenge of DG re-ID remains largely unexplored. In this work, we address this gap by applying alignment and uniformity to person re-ID, balancing feature discriminability and generalizability to learn domain-invariant representations.

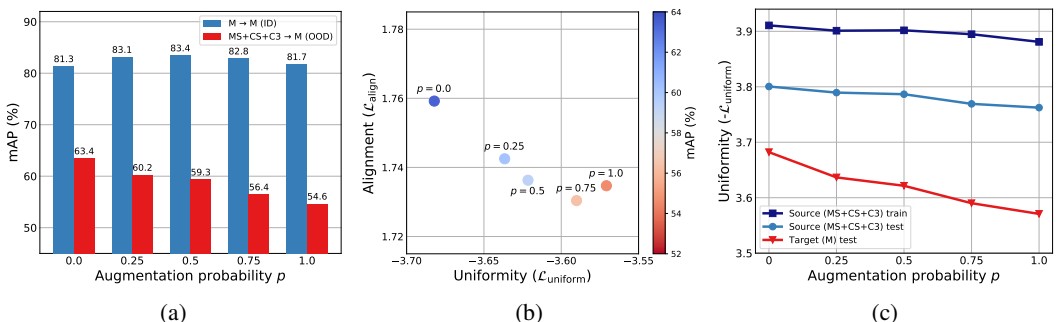

Figure 1: **Analysis on polarized effect of data augmentations on in-distribution (ID) and out-of-distribution (OOD).** (a) mAP (%) on Market-1501 of models trained on the same dataset (ID) and MS+CS+C3 (OOD) with varying augmentation probabilities. (b) Alignment ($\mathcal{L}_{align}$) and uniformity ($\mathcal{L}_{uniform}$) of OOD scenarios (MS+CS+C3 $\rightarrow$ M). Counterintuitively, augmentations lead to more alignment but less uniformity, indicating that the model fails to sufficiently preserve the diverse information from the data distribution. (c) Uniformity ($-\mathcal{L}_{uniform}$) vs. augmentation probability for the source and target datasets in MS+CS+C3 $\rightarrow$ M. Higher probabilities result in less uniformity, especially under distribution shifts, indicating an insufficiency in representing OOD data.

## 3 Method

**Problem Formulation.** Given a set of $K$ source domains, $\mathcal{D}_S = \{\mathcal{D}_k\}_{k=1}^K$, each domain $\mathcal{D}_k = \{(x_i, y_i)\}_{i=1}^{N_k}$ consists of images $x_i$ and corresponding identity labels $y_i$, where $N_k$ denotes the number of images in a source domain $\mathcal{D}_k$. Using these source domains, we train a model $f_\theta$, parameterized by $\theta$, to extract person representations $\mathbf{f}_i = f_\theta(x_i) \in \mathbb{R}^d$ from the image $x_i$, where $d$ is the dimensionality of the representation space. The trained model is then evaluated on a target domain $\mathcal{D}_T$, which is unseen during training. While a general *homogeneous* DG task has a consistent label space across source and target domains within a closed-set setting, generalizable person re-ID is a *heterogeneous* DG problem, where each domain has a disjoint label space from the others [79, 103]. Consequently, it is an open-set retrieval task where trained models need to identify unseen classes.

### 3.1 Polarized Effect of Data Augmentation on In- and Out-of-Distribution

Data augmentations, which apply random transformations to input data, are widely used across various tasks to improve training efficiency and model robustness [15, 39, 70, 89]. In the DG re-ID task, however, certain augmentations have shown *a polarized effect: they enhance retrieval performance on in-distribution data while potentially degrading it on out-of-distribution data*. For instance, Random Erasing [101], which selectively erases pixels from parts of input images, has been shown to deteriorate cross-domain re-ID performance [56]. As a result, most DG re-ID methods [10, 12, 36, 86, 94] have simply discarded this technique despite its usefulness in standard re-ID settings. Nonetheless, the underlying phenomenon of this polarized effect remains underexplored.

To investigate the polarized effect on re-ID performance between in-distribution (ID) and out-of-distribution (OOD) scenarios, we conduct experiments[1] using varying augmentation probabilities (*i.e.,* the probability of applying data augmentation to input images). For data augmentations, we employ the widely used RandAugment [11], known for its effectiveness across various vision tasks, which randomly applies transformations sampled from a predefined set, including comprehensive geometric manipulations and color variations. Considering the fine-grained domain characteristics of person re-ID, we exclude transformations that cause severe color distortion, such as Invert and Solarize, from the predefined set, and additionally utilize Random Erasing [101]. We train models with varying augmentation probabilities using a standard training pipeline that employs cross-entropy and batch-hard triplet loss [30, 56]. Following the existing DG re-ID protocol [86], the training set of MSMT17 (MS) [83], CUHK03 (C3) [44], and CUHK-SYSU (CS) [85] are used for model training as source domains, and Market-1501 (M) [95] as the target domain.

Fig. 1a compares the performance on the Market-1501 dataset of two models: one trained on the same dataset (ID) and the other trained on MS+CS+C3 (OOD). While data augmentation improves ID performance, OOD performance consistently deteriorates as the augmentation probability increases.

---

[1]More details are provided in the appendix.

This discrepancy highlights the polarized effect of data augmentations in the open-set nature of person re-ID. In closed-set recognition tasks, strong class invariance learned through augmentations can enhance generalization. However, in open-set retrieval tasks, where the model needs to handle unseen classes, learning diverse visual information becomes more crucial for generalization [58, 87]. Although augmentations improve model robustness within training distributions, they can also lead the model to focus on dominant visual information that is easily invariant to augmentations, as shown in Fig. 2. Consequently, this can result in *sparse* representation spaces, where the model focuses on learning dominant features while neglecting subtle cues that can be generalized to other domains.

To further explore the effect of data augmentations on the representation space, we leverage the concepts of *alignment* and *uniformity* [81], which are key properties of feature distributions on the unit hypersphere. Alignment is defined as the expected distance between positive pairs:

$$\mathcal{L}_{\text{align}} \triangleq \log \mathop{\mathbb{E}}_{(i,j)\sim\mathcal{P}_{\text{pos}}} [\|\mathbf{f}_i - \mathbf{f}_j\|_2^2], \quad (1)$$

where $\mathcal{P}_{\text{pos}}$ is the distribution of positive pairs. Uniformity, on the other hand, is defined by the logarithm of average pairwise Gaussian potential:

$$\mathcal{L}_{\text{uniform}} \triangleq \log \mathop{\mathbb{E}}_{(i,j)\sim\mathcal{P}_{\text{data}}} [e^{-2\|\mathbf{f}_i-\mathbf{f}_j\|_2^2}], \quad (2)$$

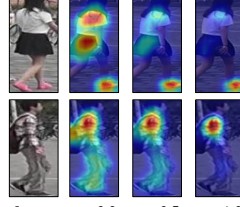

Figure 2: Grad-CAM [68] across different probabilities of data augmentations.

where $\mathcal{P}_{\text{data}}$ is the distribution of given data. These two properties reflect that positive pairs are close to each other (alignment) while the overall distribution of embeddings is uniformly spread on the hypersphere (uniformity). Several studies have demonstrated that both are essential for generalization, ensuring that the representation space achieves feature discriminability while preserving maximal information from the data [21, 65, 80].

Fig. 1b illustrates the alignment and uniformity of representations in OOD scenarios (Market-1501 for MS+CS+C3 → Market-1501) for models trained with varying augmentation probabilities. As depicted, the use of augmentations leads to more alignment, thereby enhancing intra-class invariance compared to models trained without augmentation. Conversely, in terms of uniformity, higher augmentation probabilities result in less uniform embeddings, indicating that the model fails to sufficiently preserve the diverse information from the data. However, to achieve generalizability in the open-set re-ID task, learning diverse information (*i.e., more uniformity*) is crucial [7, 12, 58]. Furthermore, as shown in Fig. 1c, the feature embeddings become increasingly less uniform with higher probabilities, and this becomes more evident as distributional shifts occur, as indicated by the red plot in the same figure. It suggests that simple training with data augmentations causes the model to become dominated by specific in-distribution data and fail to learn diverse visual cues, leading to degraded generalization performance with a sparse representation space with less uniformity.

Our analysis highlights the polarized effect of data augmentations in person re-ID, showing that while they enhance in-distribution performance, they can deteriorate out-of-distribution performance by leading to a sparse representation space. Nevertheless, data augmentations remain a promising technique to improve generalization by increasing both the diversity and robustness of training data. In the following subsection, we present a method to mitigate this effect by incorporating alignment and uniformity, using both original and augmented images to balance feature discriminability and generalizability.

### 3.2 Balancing Alignment and Uniformity

We introduce a simple yet effective framework, Balancing Alignment and Uniformity (BAU), which mitigates the polarized effect of data augmentations by maintaining a balance between alignment and uniformity. The overview is illustrated in Fig. 3. Given an input batch, we generate augmented views of the images, $\tilde{x} = t(x)$, where $t \sim \mathcal{T}$ denotes augmentations sampled from the distribution $\mathcal{T}$. Our model then extracts features from both the original and augmented images, denoted as $\mathbf{f}_i = f_\theta(x_i)$ and $\tilde{\mathbf{f}}_i = f_\theta(\tilde{x}_i)$, respectively. Since simple training with augmented images can lead to polarized effects (Sec. 3.1), we apply both alignment and uniformity losses to the features of the augmented images to achieve both feature discriminability and generalizability simultaneously. Specifically, the alignment loss enhances feature discriminability by promoting invariance to diverse augmentations, while the uniformity loss encourages generalizability by striving for a uniform distribution of features on the hypersphere, thereby preserving diverse visual information from the data.

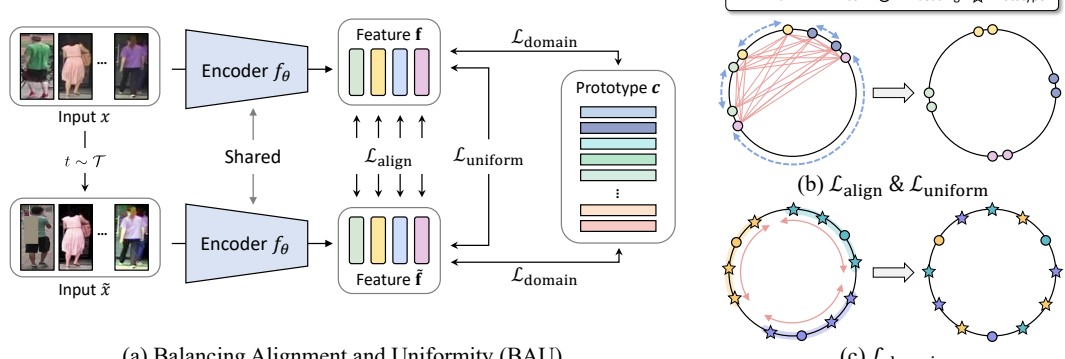

(a) Balancing Alignment and Uniformity (BAU)

(b) $\mathcal{L}_{\text{align}}$ & $\mathcal{L}_{\text{uniform}}$

(c) $\mathcal{L}_{\text{domain}}$

Figure 3: **Overview of the proposed framework. In (b) and (c), each color represents a different identity and domain, respectively.** (a) With original and augmented images, we apply alignment and uniformity losses to balance feature discriminability and generalization capability. We further introduce a domain-specific uniformity loss to mitigate domain bias. (b) $\mathcal{L}_{\text{align}}$ pulls positive features closer, while $\mathcal{L}_{\text{uniform}}$ pushes all features apart to maintain diversity. (c) $\mathcal{L}_{\text{domain}}$ uniformly distributes each domain's features and prototypes, reducing domain bias and thus enhancing generalization.

**Alignment Loss.** We reformulate the alignment loss to minimize the expected feature distance of positive pairs between original and augmented images, defined as:

$$\mathcal{L}_{\text{align}} = \frac{1}{|\mathcal{I}_{\text{pos}}|} \sum_{(i,j)\in\mathcal{I}_{\text{pos}}} \|\tilde{\mathbf{f}}_i - \mathbf{f}_j\|_2^2, \tag{3}$$

where $\mathcal{I}_{\text{pos}} = \{(i,j) \mid y_i = y_j\}$ is the index set for the positive pairs within a mini-batch, and $|\cdot|$ denotes the cardinality of the set. With alignment loss, the model can learn invariance to various transformations introduced by data augmentations, thereby enhancing feature discriminability. However, some augmented samples may suffer from significant corruption due to aggressive augmentations, and learning invariance with these samples can potentially degrade the training process.

To address this issue, we introduce a weighting strategy for the alignment loss that considers the reliability of augmented samples. Based on studies that handle noisy samples by leveraging the relationship with nearest neighbors [8, 9, 32, 84, 100], we compute the weight as the Jaccard similarity of $k$-reciprocal nearest neighbors between the augmented sample and the original sample, defined by:

$$w_{ij} = \frac{|\mathcal{R}_k(\tilde{\mathbf{f}}_i) \cap \mathcal{R}_k(\mathbf{f}_j)|}{|\mathcal{R}_k(\tilde{\mathbf{f}}_i) \cup \mathcal{R}_k(\mathbf{f}_j)|} \in [0,1], \tag{4}$$

where $\mathcal{R}_k(\mathbf{f}_i)$ is the set of indices for $k$-reciprocal nearest neighbors within a mini-batch of the feature $\mathbf{f}_i$. Intuitively, a low weight implies that the augmented feature $\tilde{\mathbf{f}}_i$ and the original feature $\mathbf{f}_j$ are not strongly correlated, indicating that learning invariance between them can provide unreliable information to each other. We reformulate the alignment loss in Eq. (3) with weights $w$, computed as:

$$\mathcal{L}_{\text{align}} = \sum_{(i,j)\in\mathcal{I}_{\text{pos}}} \bar{w}_{ij} \|\tilde{\mathbf{f}}_i - \mathbf{f}_j\|_2^2, \tag{5}$$

where $\bar{w} = \frac{w}{\sum_{(i,j)\in\mathcal{I}_{\text{pos}}} w_{ij}}$ is the normalized weight. This strategy allows the alignment loss to focus on reliable pairwise relationships between the original and augmented images without noisy samples.

**Uniformity Loss.** Following Wang and Isola [81], we compute the uniformity loss to balance feature discriminability and generalizability as:

$$\mathcal{L}_{\text{uniform}} = \log\left(\frac{1}{|\mathcal{I}_{\text{data}}|} \sum_{(i,j)\in\mathcal{I}_{\text{data}}} e^{-2\|\mathbf{f}_i-\mathbf{f}_j\|_2^2}\right) + \log\left(\frac{1}{|\mathcal{I}_{\text{data}}|} \sum_{(i,j)\in\mathcal{I}_{\text{data}}} e^{-2\|\tilde{\mathbf{f}}_i-\tilde{\mathbf{f}}_j\|_2^2}\right), \tag{6}$$

where $\mathcal{I}_{\text{data}} = \{(i,j) \mid i \neq j\}$ is the index set of all distinct pairs within a mini-batch. By incorporating the uniformity loss, the learned representations can be uniformly distributed on the

hypersphere, maintaining the diversity of the feature space. This feature diversity is crucial for the generalization capability, as it prevents the model from overfitting to sparse dominant representations and encourages learning subtle cues that can be generalized across different domains.

**Domain-specific Uniformity Loss.** While the uniformity loss within a mini-batch can improve the diversity of the feature space, batched samples may not fully capture the global structure of the entire representation space. Additionally, high uniformity alone does not guarantee domain invariance, since features from the same domain may still cluster together, as illustrated in Fig. 3 (c). To address these issues, we employ a feature memory bank, $\mathcal{M} = \{\mathbf{c}_1, ..., \mathbf{c}_N\} \in \mathbb{R}^{N \times d}$ with class prototypes $\mathbf{c} \in \mathbb{R}^d$, where $N$ is the number of classes in the given datasets. Each class prototype represents the feature vector for its respective class and is updated continuously during training using the momentum strategy [23, 28] as:

$$\mathbf{c} \leftarrow \mu \cdot \mathbf{c} + (1 - \mu) \cdot \mathbf{f}, \tag{7}$$

where $\mathbf{f}$ is the extracted features of original images with the same class label as $\mathbf{c}$, and $\mu \in [0, 1]$ is the momentum coefficient. To mitigate the inherent domain bias in simple uniformity, we additionally apply a domain-specific uniformity loss, which enhances the uniformity between the features and prototypes within their corresponding domain as follows:

$$\mathcal{L}_{\text{domain}} = \log \left( \frac{\sum_i \sum_{j \in \mathcal{N}(\mathbf{f}_i)} e^{-2\|\mathbf{f}_i - \mathbf{c}_j\|_2^2}}{\sum_i N} \right) + \log \left( \frac{\sum_i \sum_{j \in \mathcal{N}(\tilde{\mathbf{f}}_i)} e^{-2\|\tilde{\mathbf{f}}_i - \mathbf{c}_j\|_2^2}}{\sum_i N} \right), \tag{8}$$

where $\mathcal{N}(\mathbf{f})$ is the index set of *nearest prototypes of* $\mathbf{f}$ *that are from the same source domain but different class*, and $N$ is the number of nearest prototypes, which is set to match the size of the mini-batch. This loss attempts to uniformly distribute the features of each domain, reducing domain bias for domain-invariant representation space. Furthermore, by using a memory bank to compute uniformity with nearest prototypes, we can efficiently consider the overall structure of the representation space.

**Overall Training Objective.** The overall loss function of BAU is then given by:

$$\mathcal{L}_{\text{BAU}} = \mathcal{L}_{\text{ce}} + \mathcal{L}_{\text{tri}} + \lambda \mathcal{L}_{\text{align}} + \mathcal{L}_{\text{uniform}} + \mathcal{L}_{\text{domain}}, \tag{9}$$

where $\mathcal{L}_{\text{ce}}$ and $\mathcal{L}_{\text{tri}}$ are the cross-entropy loss and triplet loss, respectively, applied only to the original images. Here, $\lambda$ is the weighting parameter for the alignment loss. By ensuring alignment and uniformity with both original and augmented images, BAU effectively mitigates the detrimental effect of data augmentations while simultaneously exploiting the diversity they introduce. Moreover, BAU is a simple yet effective framework that directly regularizes the representation space without the need for advanced network architectures or complex training procedures.

## 4 Experiments

### 4.1 Datasets and Evaluation Protocols

Table 1: Statistics of the used datasets.

| Dataset | #ID | #Image | #Camera |
|---|---|---|---|
| Market1501 (M) [95] | 1,501 | 32,217 | 6 |
| MSMT17 (MS) [83] | 4,101 | 126,441 | 15 |
| CUHK02 (C2) [43] | 1,816 | 7,264 | 10 |
| CUHK03 (C3) [44] | 1,467 | 14,096 | 2 |
| CUHK-SYSU (CS) [85] | 11,934 | 34,574 | 1 |
| PRID [31] | 200 | 1,134 | 2 |
| GRID [55] | 250 | 1,275 | 8 |
| VIPeR [26] | 632 | 1,264 | 2 |
| iLIDs [97] | 119 | 476 | 2 |

Table 2: Evaluation protocols.

| Setting | Training Data | Testing Data |
|---|---|---|
| Protocol-1 | Full-(M+C2+C3+CS) | PRID, GRID, VIPeR, iLIDs |
| Protocol-2 | M+MS+CS
M+CS+C3
MS+CS+C3 | C3
MS
M |
| Protocol-3 | Full-(M+MS+CS)
Full-(M+CS+C3)
Full-(MS+CS+C3) | C3
MS
M |

We conduct experiments using the following datasets: Market-1501 [95], MSMT17 [83], CUHK02 [43], CUHK03 [44], CUHK-SYSU [85], PRID [31], GRID [55], VIPeR [26], and iLIDs [97], with dataset statistics shown in Table 1. For simplicity, we denote Market-1501, MSMT17, CUHK02, CUHK03, and CUHK-SYSU as M, MS, C2, C3, and CS, respectively. Evaluation metrics include mean average precision (mAP) and cumulative matching characteristic (CMC) at Rank-1.

Following previous studies [71, 86, 90, 94], we evaluate the proposed method across three protocols, as detailed in Table 2. For Protocol-1, we utilize all images from M, C2, C3, and CS, including both training and testing data, for model training. We then evaluate the model on four small-scale re-ID datasets, specifically PRID, GRID, VIPeR, and iLIDs. The final performance on these small-scale

Table 3: Comparison with state-of-the-art methods on Protocol-1. Since DukeMTMC-reID [98], denoted as D in the table, has been withdrawn, it is not utilized for our training.

| Source | Method | PRID | | GRID | | VIPeR | | iLIDs | | Average | |
|---|---|---|---|---|---|---|---|---|---|---|---|
| | | mAP | Rank-1 | mAP | Rank-1 | mAP | Rank-1 | mAP | Rank-1 | mAP | Rank-1 |
| M+D +C2+C3 +CS | DIMN [71] | 52.0 | 39.2 | 41.1 | 29.3 | 60.1 | 51.2 | 78.4 | 70.2 | 57.9 | 47.5 |
| | DualNorm [34] | 64.9 | 60.4 | 45.7 | 41.4 | 58.0 | 53.9 | 78.5 | 74.8 | 61.8 | 57.6 |
| | SNR [36] | 66.5 | 52.1 | 47.7 | 40.2 | 61.3 | 52.9 | 89.9 | 84.1 | 66.4 | 57.3 |
| | DDAN [6] | 67.5 | 62.9 | 50.9 | 46.2 | 60.8 | 56.5 | 81.2 | 78.0 | 65.1 | 60.9 |
| | RaMoE [12] | 67.3 | 57.7 | 54.2 | 46.8 | 64.6 | 56.6 | 90.2 | 85.0 | 62.0 | 61.5 |
| | DMG-Net [4] | 68.4 | 60.6 | 56.6 | 51.0 | 60.4 | 53.9 | 83.9 | 79.3 | 67.3 | 61.2 |
| | GDNorm [53] | 79.9 | 72.6 | 63.8 | 55.4 | 74.1 | 66.1 | 87.2 | 81.3 | 76.3 | 68.9 |
| | DTIN [35] | 79.7 | 71.0 | 60.6 | 51.8 | 70.7 | 62.9 | 87.2 | 81.8 | 74.6 | 66.9 |
| | StyCon [48] | 78.9 | 71.0. | 60.4 | 50.7 | 74.4 | 66.8 | 86.9 | 80.7 | 75.2 | 67.3 |
| M +C2+C3 +CS | QAConv$_{50}$ [50] | 62.2 | 52.3 | 57.4 | 48.6 | 66.3 | 57.0 | 81.9 | 75.0 | 67.0 | 58.2 |
| | M$^3$L [94] | 65.3 | 55.0 | 50.5 | 40.0 | 68.2 | 60.8 | 74.3 | 65.0 | 64.6 | 55.2 |
| | MetaBIN [10] | 70.8 | 61.2 | 57.9 | 50.2 | 64.3 | 55.9 | 82.7 | 74.7 | 68.9 | 60.5 |
| | META [86] | 71.7 | 61.9 | 60.1 | 52.4 | 68.4 | 61.5 | 83.5 | 79.2 | 70.9 | 63.8 |
| | ACL [90] | 73.4 | 63.0 | 65.7 | 55.2 | **75.1** | **66.4** | 86.5 | 81.8 | 75.2 | 66.6 |
| | StyCon [48] | 78.1 | 69.7 | 62.1 | 53.4 | 71.2 | 62.8 | 84.8 | 78.0 | 74.1 | 66.0 |
| | **BAU (Ours)** | **77.2** | **68.4** | **68.1** | **59.8** | 74.6 | 66.1 | **88.7** | **83.7** | **77.2** | **69.5** |

Table 4: Comparison with state-of-the-art methods on Protocol-2 and Protocol-3.

| Setting | Method | M+MS+CS → C3 | | M+CS+C3 → MS | | MS+CS+C3 → M | | Average | |
|---|---|---|---|---|---|---|---|---|---|
| | | mAP | Rank-1 | mAP | Rank-1 | mAP | Rank-1 | mAP | Rank-1 |
| Protocol-2 | SNR [36] | 8.9 | 8.9 | 6.8 | 19.9 | 34.6 | 62.7 | 16.8 | 30.5 |
| | QAConv$_{50}$ [50] | 25.4 | 24.8 | 16.4 | 45.3 | 63.1 | 83.7 | 35.0 | 51.3 |
| | M$^3$L [94] | 34.2 | 34.4 | 16.7 | 37.5 | 61.5 | 82.3 | 37.5 | 51.4 |
| | MetaBIN [10] | 28.8 | 28.1 | 17.8 | 40.2 | 57.9 | 80.1 | 34.8 | 49.5 |
| | META [86] | 36.3 | 35.1 | 22.5 | 49.9 | 67.5 | 86.1 | 42.1 | 57.0 |
| | ACL [90] | 41.2 | 41.8 | 20.4 | 45.9 | 74.3 | 89.3 | 45.3 | 59.0 |
| | **BAU (Ours)** | **42.8** | **43.9** | **24.3** | **50.9** | **77.1** | **90.4** | **48.1** | **61.7** |
| Protocol-3 | SNR [36] | 17.5 | 17.1 | 7.7 | 22.0 | 52.4 | 77.8 | 25.9 | 39.0 |
| | QAConv$_{50}$ [50] | 32.9 | 33.3 | 17.6 | 46.6 | 66.5 | 85.0 | 39.0 | 55.0 |
| | M$^3$L [94] | 35.7 | 36.5 | 17.4 | 38.6 | 62.4 | 82.7 | 38.5 | 52.6 |
| | MetaBIN [10] | 43.0 | 43.1 | 18.8 | 41.2 | 67.2 | 84.5 | 43.0 | 56.3 |
| | META [86] | 47.1 | 46.2 | 24.4 | 52.1 | 76.5 | 90.5 | 49.3 | 62.9 |
| | ACL [90] | 49.4 | 50.1 | 21.7 | 47.3 | 76.8 | 90.6 | 49.3 | 62.7 |
| | **BAU (Ours)** | **50.6** | **51.8** | **26.8** | **54.3** | **79.5** | **91.1** | **52.3** | **65.7** |

datasets is obtained by averaging the results of 10 repeated random splits of the query and gallery sets. For Protocol-2 and Protocol-3, we follow a leave-one-out evaluation setting with four large-scale datasets: M, MS, C3, and CS. Three datasets are used as the source domain, and the remaining one is used as the target domain. Protocol-2 uses only the training data from the source domains for model training, whereas Protocol-3 utilizes both the training and testing data from the source domains. Since CUHK-SYSU (CS) is a person search dataset with a single camera view, it is only used for training.

## 4.2 Implementation Details

Following previous studies [34, 50, 51, 86, 90], we use ResNet-50 [29] pre-trained on ImageNet [13] with instance normalization layers as our backbone. All images are resized to $256 \times 128$. For each iteration, we sample 256 images, consisting of 64 identities with 4 instances for each identity. The total batch size during training is 512, including both original and augmented images. Random flipping, cropping, erasing [101], RandAugment [11], and color jitter are used for data augmentation. We train the model for 60 epochs using Adam [38] with a weight decay of $5 \times 10^{-4}$. The initial learning rate is set to $3.5 \times 10^{-4}$ and is decreased by a factor of 10 at the 30th and 50th epochs. A warmup strategy is applied during the first 10 epochs. The momentum $\mu$ is set to 0.1. We empirically set the weighting parameter $\lambda$ to 1.5 and $k$ for the weighting strategy to 10. We implement our framework in PyTorch [64] and utilize two RTX-3090 GPUs for training.

## 4.3 Comparison with State-of-the-Arts

We compare our method with state-of-the-art DG re-ID methods on Protocol-1, with the results presented in Table 3. We also report the results of previous studies that use DukeMTMC-reID [98] for training, denoted as D in the table, while we exclude this dataset from training since it has been

Table 5: Ablation study of loss functions for augmented images.

| Method | M+MS+CS → C3 | | M+CS+C3 → MS | | MS+CS+C3 → M | | Average | |
|---|---|---|---|---|---|---|---|---|
| | mAP | Rank-1 | mAP | Rank-1 | mAP | Rank-1 | mAP | Rank-1 |
| Baseline (w/o augmented images) | 39.2 | 38.9 | 18.9 | 44.8 | 68.3 | 86.3 | 42.1 | 56.7 |
| $\mathcal{L}_{ce}$ | 32.1 | 31.9 | 18.0 | 41.7 | 63.1 | 83.8 | 37.7 | 52.5 |
| $\mathcal{L}_{align}$ | 41.8 | 42.2 | 23.1 | 47.4 | 73.8 | 87.4 | 46.2 | 59.0 |
| $\mathcal{L}_{align} + \mathcal{L}_{uniform}$ | 46.9 | 47.4 | 25.3 | 51.1 | 78.1 | 89.5 | 50.1 | 62.7 |
| $\mathcal{L}_{align} + \mathcal{L}_{uniform} + \mathcal{L}_{domain}$ | **50.6** | **51.8** | **26.8** | **54.3** | **79.5** | **91.1** | **52.3** | **65.7** |

Table 6: Ablation study of the weighting strategy and the domain-specific uniformity loss.

| Weighting Strategy of $\mathcal{L}_{align}$ | Domain-specific Prototype of $\mathcal{L}_{domain}$ | M+MS+CS → C3 | | M+CS+C3 → MS | | MS+CS+C3 → M | | Average | |
|---|---|---|---|---|---|---|---|---|---|
| | | mAP | Rank-1 | mAP | Rank-1 | mAP | Rank-1 | mAP | Rank-1 |
| | | 46.2 | 46.6 | 23.9 | 49.3 | 76.7 | 90.0 | 48.9 | 62.0 |
| ✓ | | 47.1 | 47.4 | 24.7 | 50.9 | 78.3 | 88.9 | 50.0 | 62.4 |
| | ✓ | 49.5 | 51.0 | 25.1 | 51.6 | 78.6 | 90.8 | 51.1 | 64.5 |
| ✓ | ✓ | **50.6** | **51.8** | **26.8** | **54.3** | **79.5** | **91.1** | **52.3** | **65.7** |

taken down. As shown in the table, although our method utilizes fewer datasets for training, it surpasses prior state-of-the-art methods in average mAP/Rank-1 performance across the four datasets. Comparisons of the proposed method with state-of-the-art methods on Protocol-2 and Protocol-3 are presented in Table 4. The results demonstrate that our method outperforms other methods, confirming the generalization capability of BAU on large-scale datasets. Specifically, we achieve higher average mAP scores across three datasets than the previous state-of-the-art ACL [90], with improvements of +2.7% and +3.4% on protocol-2 and protocol-3, respectively.

It is also noteworthy that our method achieves state-of-the-art performance without employing advanced feature normalization modules [10, 35, 36, 48, 53] or domain-specific network architectures [12, 86, 90]. Specifically, BAU is a simple yet effective framework that regularizes the representation space by ensuring alignment and uniformity between original and augmented images, without relying on domain-adversarial [6] training or meta-learning [4, 94] strategies.

### 4.4 Ablation Study and Analysis

To evaluate the effectiveness of each component in BAU and validate our design choices, we conduct extensive ablation studies and analyses on Protocol-3, the most scalable setting in our experiments.

**Ablation study of loss functions for augmented images.** We first analyze the impact of applying different loss functions to the augmented images, as shown in Table 5. The baseline model, trained using cross-entropy and triplet losses without any augmented images (when $p = 0$ of Sec. 3.1), serves as a reference point. Applying only the cross-entropy loss $\mathcal{L}_{ce}$ to the augmented images leads to a performance drop, aligning with our analysis in Sec. 3.1 that naive training with augmented images can degrade generalization performance. This result also aligns with studies demonstrating that cross-entropy loss can easily overfit to biases and noise in the data [25, 59, 92]. When applying the proposed alignment loss $\mathcal{L}_{align}$, the performance improves, surpassing the baseline. Further incorporating the uniformity loss $\mathcal{L}_{uniform}$ significantly boosts the performance, confirming the importance of balancing both alignment and uniformity to achieve better generalization. Finally, integrating the domain-specific uniformity loss $\mathcal{L}_{domain}$ achieves the best performance, with notable improvements of +10.2%/+8.0% in average mAP/Rank-1 over the baseline. This validates that mitigating domain bias by uniformly distributing features within each domain is effective. In summary, these results highlight that while simply training with augmentations can be detrimental, carefully regularizing the representation space through alignment and uniformity allows the model to benefit from the augmented data, leading to improved generalization.

**Ablation study of weighting strategy and domain-specific uniformity loss.** We conduct an ablation study to validate the effectiveness of the proposed weighting strategy for the alignment loss and the domain-specific uniformity loss, as shown in Table 6. Without the weighting strategy in Eq. 3, the performance drops compared to the full BAU model, demonstrating the benefit of focusing on reliable pairs between original and augmented images to learn robust features. To investigate the importance of the domain-specific uniformity loss, we apply a simple uniformity loss with a feature memory bank without considering each domain (*i.e.,* modifying Eq. (8) with nearest prototypes in

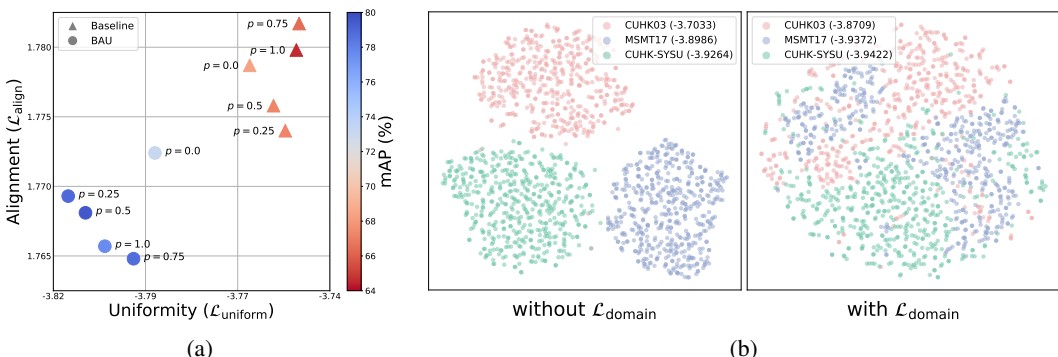

Figure 4: **Analysis of alignment and uniformity.** (a) Alignment ($\mathcal{L}_{\mathrm{align}}$) and uniformity ($\mathcal{L}_{\mathrm{uniform}}$) on Market-1501 when MS+CS+C3 $\rightarrow$ M under Protocol-3 with varying augmentation probabilities. (b) T-SNE visualization with and without the domain-specific uniformity loss $\mathcal{L}_{\mathrm{domain}}$. The values in parentheses in each legend label indicate the uniformity of the corresponding domain.

the entire domain, not intra-domain). The performance decreases, highlighting the effectiveness of promoting uniformity within each domain for learning domain-invariant features. When both the weighting strategy and domain-specific prototypes are removed, the performance drops further, showing that each component provides complementary benefits, and integrating both in the BAU framework results in better generalization. This analysis validates the effectiveness of the proposed techniques, enabling BAU to achieve strong performance in the DG re-ID task by focusing on reliable pairs in alignment and promoting uniformity within each domain.

**Analysis of alignment and uniformity.** To further validate the effect of our proposed method on the learned representation space, we analyze the alignment and uniformity properties of the baseline and BAU. Fig. 4a illustrates these properties for the representations on the testing data of Market-1501 when MS+CS+C3 $\rightarrow$ M under Protocol-3. When data augmentations are applied, the representations of the baseline become less uniform, resulting in decreased generalization performance, as discussed in Sec. 3.1. In contrast, BAU consistently maintains better alignment and uniformity compared to the baseline across all augmentation probabilities. Notably, even when data augmentation is not applied (*i.e.*, $p = 0$), BAU still outperforms the baseline. This demonstrates that in addition to preventing the polarizing effect of data augmentation, alignment and uniformity are important factors for generalization, aligning with recent studies [21, 65, 80]. In summary, by explicitly regularizing the representation space with both original and augmented images, BAU achieves better generalization performance by balancing feature discriminability and generalizability, confirming that it effectively leverages the diversity from data augmentations.

**Analysis of weighting strategy and domain-specific uniformity loss.** To explore the effect of the domain-specific uniformity loss, we visualize the t-SNE [74] plot of the training data when MS+CS+C3 $\rightarrow$ M under Protocol-3, with and without the domain-specific uniformity loss $\mathcal{L}_{\mathrm{domain}}$. As shown in Fig. 4b, without $\mathcal{L}_{\mathrm{domain}}$, the learned features exhibit domain-specific clusters, indicating a lack of domain invariance. On the other hand, applying $\mathcal{L}_{\mathrm{domain}}$ results in a more uniform distribution of features across domains, as evidenced by the increased uniformity values for each domain (reported in parentheses). This demonstrates the effectiveness of promoting uniformity within each domain to learn domain-invariant representations.

To further analyze the proposed weighting strategy for the alignment loss, we conduct quantitative comparisons across varying augmentation probabilities, with and without the weighting strategy. As shown in Fig. 5a, our weighting strategy consistently improves performance across different augmentation probabilities, with the gap becoming more pronounced at higher probabilities where severe corruption from augmentations is more likely. Specifically, at an augmentation probability of 0.5, the weighting strategy improves the mAP from 78.3% to 79.5%, and at a probability of 1.0, the improvement is even more substantial, from 66.1% to 76.1%. These results demonstrate that BAU effectively enables learning invariance with reliable augmented samples, thanks to the proposed weighting strategy, which focuses on reliable pairs between original and augmented images.

For qualitative analysis, Fig. 5b illustrates the weight scores $w$ in Eq. (4) for the alignment loss. The top row shows the original images, while the bottom row represents their augmented versions

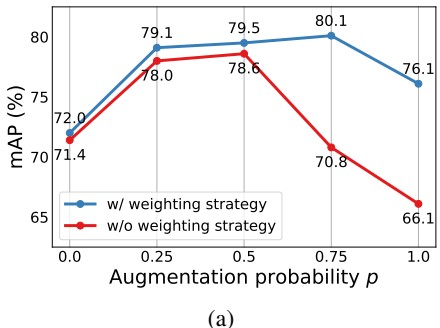
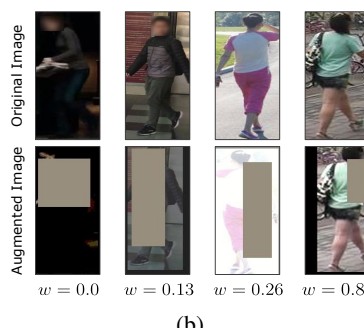

(a)                    (b)

Figure 5: **Analysis of the weighting strategy.** (a) Quantitative comparison of mAP (%) across varying augmentation probabilities, with and without the weighting strategy, on MS+CS+C3 → M under Protocol-3. The weighting strategy consistently improves performance, especially at higher augmentation probabilities, where the mAP drops significantly without it. (b) Qualitative analysis of the weight score $w$ for different pairs of original and augmented images.

with corresponding weight scores. As $w$ progressively increases from left to right, the augmented images exhibit less distortion, indicating more reliable samples with informative augmentations. The weighting strategy allows the alignment loss to focus on augmented samples that are semantically similar to the original images, facilitating invariance learning from informative augmentations.

# 5 Discussion

Although our method has shown promising results, it still has limitations to overcome. As a straight-forward approach primarily based on data augmentations for given input data, our method could face challenges under very large domain shifts. While we adopt standard image-level augmentations, we do not incorporate more advanced techniques such as adversarial data augmentations [69, 75, 99, 104] or feature-level augmentations [37, 46, 106], both of which have shown promise for generalization. Integrating these techniques with the proposed BAU framework could potentially further enhance generalization performance and would be an interesting direction for future research.

Despite these limitations, our work is the first to thoroughly investigate and address the polarized effect of data augmentations in DG re-ID. It demonstrates the significant potential of balancing alignment and uniformity to improve generalization in person re-identification. We believe that our findings could inspire further research aimed at advancing generalization capabilities in this field.

# 6 Conclusion

In this paper, we investigated the polarized effect of data augmentations in domain generalizable person re-identification. Our findings revealed that while augmentations enhance in-distribution performance, they can also lead to sparse representation spaces and deteriorate out-of-distribution performance. To address this issue, we proposed a simple yet effective framework, Balancing Alignment and Uniformity (BAU), which regularizes the representation space by maintaining a balance between alignment and uniformity. Comprehensive experiments on various benchmarks and protocols demonstrated that BAU achieves state-of-the-art performance without requiring advanced network architectures or complex training procedures. Furthermore, our extensive ablation studies and analyses validated the effectiveness of each component in BAU, emphasizing the importance of balancing alignment and uniformity to achieve robust generalization in person re-identification.

## Acknowledgements

We would like to thank the anonymous reviewers for their constructive comments and suggestions. This work was supported by the Institute of Information & communications Technology Planning & Evaluation (IITP) grant (No. RS-2023-00237965, Recognition, Action and Interaction Algorithms for Open-world Robot Service) and the National Research Foundation of Korea (NRF) grant (No. RS-2023-00208506 (2024)), both funded by the Korea government (MSIT). Prof. Sung-Eui Yoon is a corresponding author (e-mail: sungeui@kaist.edu).

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

# A    Additional Implementation Details

This section provides additional implementation details for the experiments in Sec. 3.1. RandAugment [11] includes comprehensive geometric manipulations and color variations[2]: 'AutoContrast', 'Equalize', 'Invert', 'Rotate', 'Posterize', 'Solarize', 'Color', 'Contrast', 'Brightness', 'Sharpness', 'ShearX', 'ShearY', 'TranslateX', 'TranslateY', 'Cutout', 'SolarizeAdd'. Since person re-identification is a fine-grained task that requires distinguishing individuals with subtle distinctions, severe color distortion can deteriorate feature discriminability. Therefore, we exclude 'Invert', 'Posterize', 'Solarize', and 'SolarizeAdd' from the predefined set of RandAugment and additionally utilize Random Erasing. Following the conventional training pipeline [56], we configured the in-distribution model training with a batch size of 64, consisting of 16 identities with 4 instances for each identity. We train the model for 120 epochs using Adam [38] optimizer with weight decay of $5 \times 10^{-4}$. The initial learning rate is set to $3.5 \times 10^{-4}$ and is decreased by a factor of 10 at the 40th and 70th epochs. A warmup strategy is also applied during the first 10 epochs. For the out-of-distribution model training, we sample 256 images, consisting of 64 identities with 4 instances for each identity. Other settings are the same as the implementation details in the main paper.

# B    Additional Experimental Results

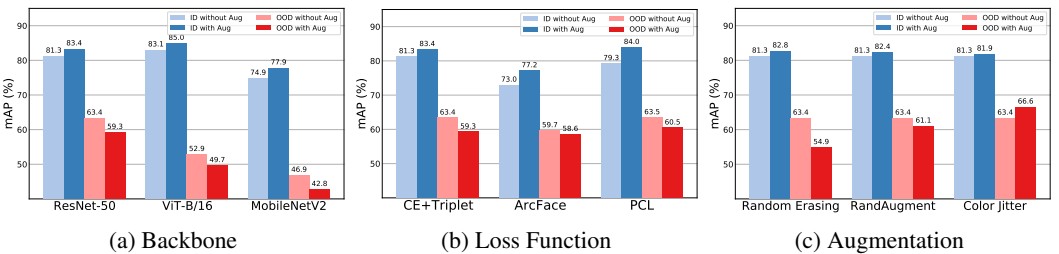

(a) Backbone                              (b) Loss Function                              (c) Augmentation

Figure 6: **Analysis on polarized effect across different types of (a) backbones, (b) loss functions, and (c) augmentations.** The experimental configurations are the same in Fig. 1 of the main paper.

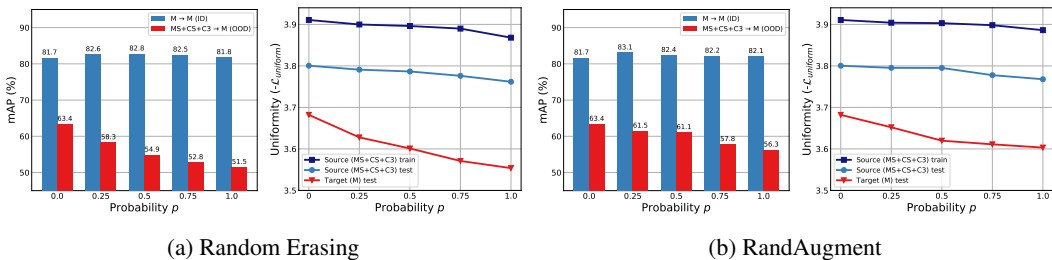

(a) Random Erasing                                          (b) RandAugment

Figure 7: **Analysis on polarized effect of (a) Random Erasing and (b) RandAugment** across augmentation probabilities. The experimental configurations are the same in Fig. 1 of the main paper.

## B.1    Commonality Analysis of Polarized Effect

To further explore the commonality of the polarized effect in data augmentation for DG re-ID, we investigate its presence across different backbones, loss functions, and augmentation types.

**Backbone and loss function.**    We examine the polarized effect across different backbone architectures, including transformer-based (ViT-B/16 [16]) and lightweight (MobileNetV2 [67]) networks. Following the same experimental protocol outlined in Sec. 3.1, we train these models using cross-entropy and batch-hard triplet loss[30], with and without data augmentations (*i.e.*, Random Erasing [101] and RandAugment [11]). As shown in Fig 6a, the polarized effect is consistently observed across different backbones. Additionally, we investigate this effect across different loss functions on a ResNet-50 backbone, including ArcFace [14] and PCL [23, 42], which are widely used for re-ID and

[2]https://github.com/tensorflow/tpu/tree/master/models/official/efficientnet

Table 7: Evaluation of BAU with other backbones and loss functions on Protocol-2.

| Method | M+MS+CS → C3 | | M+CS+C3 → MS | | MS+CS+C3 → M | | Average | |
|---|---|---|---|---|---|---|---|---|
| | mAP | Rank-1 | mAP | Rank-1 | mAP | Rank-1 | mAP | Rank-1 |
| *BAU with other backbones* | | | | | | | | |
| MobileNetV2 | 21.4 | 20.2 | 10.1 | 24.3 | 46.9 | 69.4 | 26.1 | 38.0 |
| **+ BAU (Ours)** | **27.5** | **27.4** | **12.9** | **31.3** | **59.1** | **78.5** | **33.2** | **45.7** |
| ViT-B/16 | 31.8 | 30.9 | 14.8 | 29.2 | 52.9 | 74.6 | 33.2 | 44.9 |
| **+ BAU (Ours)** | **37.3** | **36.9** | **20.0** | **40.5** | **63.5** | **80.6** | **40.3** | **52.7** |
| *BAU with other loss functions* | | | | | | | | |
| ArcFace | 33.8 | 33.9 | 17.6 | 37.6 | 58.9 | 79.5 | 36.8 | 50.3 |
| **+ BAU (Ours)** | **37.8** | **38.3** | **23.5** | **50.8** | **72.5** | **88.7** | **44.6** | **59.3** |
| PCL | 34.6 | 35.5 | 16.4 | 33.7 | 63.5 | 83.4 | 38.2 | 50.9 |
| **+ BAU (Ours)** | **39.7** | **40.9** | **21.8** | **47.0** | **74.3** | **88.4** | **45.3** | **58.8** |

retrieval tasks. As shown in Fig. 6b, the polarized effect persists across all loss functions, and these results confirm that the polarized effect is not limited to specific architectures and loss functions.

**Augmentation.** We investigate the existence of the polarized effect of individual augmentations used in BAU, specifically Random Erasing, RandAugment, and Color Jitter. Following the experimental setting outlined in Sec. 3.1, we conduct experiments using a ResNet-50 backbone with cross-entropy and batch-hard triplet loss. As shown in Fig. 6c, the results indicate that the performance drop in the unseen domain is primarily driven by the polarized effects observed in Random Erasing and RandAugment, while Color Jitter does not exhibit this behavior, consistent with previous findings in the field. While Random Erasing and RandAugment can introduce significant distortions (*e.g.*, pixel drops) to images, Color Jitter only provides simpler color distortions, which could enhance model robustness to variations in lighting and color conditions in unseen environments. Additionally, as shown in Fig. 7, increasing the augmentation probability of Random Erasing and RandAugment further degrades performance and reduces uniformity, confirming the polarized effect caused by these augmentations in DG Re-ID.

In summary, we consistently observe the polarized effect across different backbones, loss functions, and augmentation types, suggesting that this phenomenon is general in DG re-ID. Furthermore, the proposed BAU consistently improves various baselines with different backbones and loss functions (Table. 7) and various augmentation types (Table. 8).

## B.2 Evaluation of BAU across Backbones and Loss Functions

To further validate the versatility and effectiveness of the proposed BAU framework, we conduct experiments across different backbone architectures and loss functions.

We first apply BAU to two distinct types of backbones: MobileNetV2 [67], a lightweight network, and ViT-B/16 [16], a transformer-based architecture. For training, we use cross-entropy and batch-hard triplet loss [30] with Random Erasing [101] and RandAugment [11] for augmentations. As shown in Table. 7, BAU consistently improves the performance of both baseline models across different backbones. For example, with MobileNetV2, BAU enhances the average mAP from 26.1% to 33.2% and Rank-1 from 38.0% to 45.7%, representing significant improvements. These results demonstrate the broad applicability of BAU across different backbones, suggesting that the proposed framework is effective regardless of underlying architecture.

We further evaluate the effectiveness of BAU on models trained with different loss functions, Arc-Face [14] and PCL [23, 42]. For training, we employ a ResNet-50 backbone with Random Erasing and RandAugment for augmentations. As shown in Table. 7, BAU consistently improves the performance of both baselines across different loss functions. For instance, with ArcFace, BAU increases the average mAP from 36.8% to 44.6% and Rank-1 from 50.3% to 59.3%, delivering substantial gains. These results confirm that BAU is not limited to any specific loss function and can be seamlessly integrated into various re-ID loss formulations to enhance generalization capabilities.

In summary, these results validate the effectiveness and versatility of BAU, showing that it can be applied across diverse architectures and loss functions to improve generalization in DG re-ID. Notably, BAU achieves these improvements as a simple regularization technique, without requiring

Table 8: Ablation study of data augmentations on Protocol-3.

| Random Erasing | RandAugment | Color Jitter | M+MS+CS → C3 | | M+CS+C3 → MS | | MS+CS+C3 → M | | Average | |
|---|---|---|---|---|---|---|---|---|---|---|
| | | | mAP | Rank-1 | mAP | Rank-1 | mAP | Rank-1 | mAP | Rank-1 |
| - | - | - | 43.1 | 42.8 | 23.1 | 47.2 | 72.0 | 87.1 | 46.1 | 59.0 |
| ✓ | - | - | 44.3 | 45.5 | 23.3 | 47.4 | 74.1 | 88.8 | 47.2 | 60.6 |
| - | ✓ | - | 47.3 | 47.1 | 24.4 | 51.3 | 77.1 | 89.9 | 49.6 | 62.8 |
| - | - | ✓ | 47.2 | 47.6 | 25.0 | 50.9 | 78.0 | 90.5 | 50.1 | 63.0 |
| ✓ | ✓ | - | 48.1 | 47.7 | 25.3 | 51.7 | 78.7 | 90.1 | 50.7 | 63.2 |
| ✓ | - | ✓ | 47.9 | 48.5 | 25.9 | 51.9 | 78.4 | 90.6 | 50.7 | 63.7 |
| - | ✓ | ✓ | 49.4 | 50.2 | 26.3 | 53.3 | 79.0 | 90.4 | 51.6 | 64.6 |
| ✓ | ✓ | ✓ | **50.6** | **51.8** | **26.8** | **54.3** | **79.5** | **91.1** | **52.3** | **65.7** |

Table 9: Ablation study of applying probabilities of data augmentations.

| $p$ | M+MS+CS → C3 | | M+CS+C3 → MS | | MS+CS+C3 → M | | Average | |
|---|---|---|---|---|---|---|---|---|
| | mAP | Rank-1 | mAP | Rank-1 | mAP | Rank-1 | mAP | Rank-1 |
| 0.0 | 43.1 | 42.8 | 23.1 | 47.2 | 72.0 | 87.1 | 46.1 | 59.0 |
| 0.25 | 48.2 | 48.6 | **27.0** | **55.0** | 79.1 | 90.9 | 51.4 | 64.8 |
| 0.5 | **50.6** | **51.8** | 26.8 | 54.3 | 79.5 | 91.1 | **52.3** | **65.7** |
| 0.75 | 49.6 | 50.1 | 25.1 | 51.5 | **80.1** | **91.8** | 51.6 | 64.5 |
| 1.0 | 47.9 | 47.6 | 24.0 | 49.1 | 76.1 | 89.1 | 49.3 | 61.9 |

complex training procedures or additional trainable parameters. This highlights its efficiency and potential for easy integration into various baseline models in the DG re-ID task.

## B.3 Ablation Study of Data Augmentation

Table 8 presents an ablation study on the impact of different data augmentations on the generalization performance of our method under Protocol-3. We investigate three augmentations used in the proposed method: Random Erasing [101], RandAugment [11], and Color Jitter. As shown in the table, the results demonstrate that each augmentation technique improves the generalization performance, indicating that our method successfully mitigates the polarized effect of data augmentations. Specifically, even though Random Erasing has generally been shown to degrade generalization performance in previous studies, our method can effectively exploit the advantages of this augmentation technique. Furthermore, it is worth noting that even without any augmentations, the proposed method still outperforms the baseline (see Table 5 in the main paper), highlighting the effectiveness of balancing alignment and uniformity in improving generalization. When all three augmentations are applied together, we achieve the best performance, with an average mAP of 52.3% and Rank-1 accuracy of 65.7%. This highlights that the diversity of data augmentations enables the proposed method to learn more diverse information from the data, thereby enhancing feature generalizability.

To further investigate the impact of data augmentation on our method, we conduct experiments by varying the probability $p$ of applying data augmentations during training. As shown in Table 9, increasing the augmentation probability generally leads to better generalization performance up to a certain point. Setting $p$ to 0.5 yields the best results, so we empirically set this configuration. Further increasing the probability to 0.75 or 1.0 leads to a slight decrease in performance, indicating that excessive overlap of the three augmentations may introduce noise and hinder the learning process.

In summary, these experimental results validate the effectiveness of data augmentations in enhancing the generalization capability for person re-identification – a potential that previous studies have not fully explored. The proposed BAU framework enables the model to leverage the diversity introduced by augmentations, resulting in more robust and generalizable representations for unseen domains.

## B.4 Parameter Analysis

We conduct a parameter analysis to investigate the impact of the weighting strategy and the alignment loss on the generalization performance of our proposed method. Specifically, we evaluate the effect of varying the number of k-reciprocal nearest neighbors $k$ for the weighting strategy and the weighting parameter $\lambda$ for the alignment loss on the MS+C3+CS → M setting under Protocol-3.

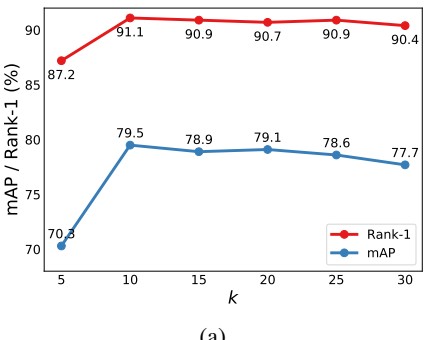 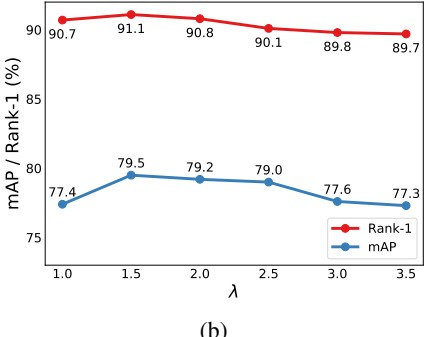

|     (a)     |     (b)     |

Figure 8: **Parameter analysis of $k$ and $\lambda$ on MS+C3+CS $\rightarrow$ M under Protocol-3.** (a) mAP/Rank-1 (%) with varying $k$-reciprocal nearest neighbors for the weighting strategy. (b) mAP/Rank-1 (%) with varying the weighting parameter $\lambda$ for the alignment loss.

Figure 8a shows the mAP and Rank-1 accuracy of our method with different values of $k$ for the weighting strategy. As $k$ increases, the performance initially improves, reaching the best results at $k = 10$ with an mAP of 79.5% and a Rank-1 accuracy of 91.1%. However, when $k$ is set to 5, the performance slightly lags behind the baseline (see Table 5 in the main paper). This result implies that the weighting strategy is not effective if unreliable weights are used. Setting $k$ to 5 is too strict for computing reliable weights, as we observed that it causes the scores of too many samples to be close to zero, leading to insufficient learning of alignment. Figure 8b presents the mAP and Rank-1 accuracy of our method with different values of $\lambda$ for the alignment loss. We observe that setting $\lambda$ to 1.5 yields the best performance. This demonstrates the importance of balancing the contribution of the alignment loss with other loss terms. When $\lambda$ is too small, the alignment loss may not sufficiently enforce feature invariance to augmentations. On the other hand, a large $\lambda$ may overemphasize alignment and hinder uniformity, leading to the learning of less diverse information.

In summary, these results highlight the importance of a reliable weighting strategy and achieving a balance between alignment and uniformity. The optimal values of $k = 10$ and $\lambda = 1.5$ are used in all our experiments reported in the main paper.

## C Broader Impacts

The insights behind our method, balancing alignment and uniformity to induce feature diversity, could be valuable for improving generalization in other fine-grained and open-set retrieval tasks. By extending this approach to related domains, such as vehicle re-identification or face recognition, the robustness and real-world applicability of various computer vision systems could be enhanced. Furthermore, by learning more generalizable domain-invariant features, the proposed method could mitigate biases (*e.g.*, race, gender) in person re-identification systems by focusing on domain-agnostic visual cues rather than spurious correlations. Further research specifically evaluating fairness impacts would be valuable to verify and quantify this potential benefit.

However, person re-identification techniques can potentially have negative impacts, such as the infringement of privacy due to the abuse of surveillance systems. For instance, these methods can raise privacy concerns, as individuals may be tracked without consent as they move through public spaces. Researchers and users of person re-identification technology should be attentive to using it in an appropriate manner while considering ethical issues. Particular care should be taken to avoid using datasets with known ethical concerns for research. The development and deployment of person re-identification systems should be accompanied by appropriate safeguards and regulations to prevent misuse and protect individual privacy rights.

