# OpenReview forum: "Generalizable Person Re-identification via Balancing Alignment and Uniformity"
_NeurIPS.cc/2024/Conference — NeurIPS 2024 poster_

### Official Review · Reviewer_4LRb · 2024-07-10

**Soundness:** 3
**Presentation:** 4
**Contribution:** 3
**Rating:** 6
**Confidence:** 4

**Summary:**

This paper investigates the side effects of data augmentation in domain generalizable person ReID problem and proposes a framework for mitigating the negative effects. It is found that the data augmentation enhances the performance of a ReID model on its training domain, while degrading the performance of it on unseen domains. To alleviate it, a simple framework, Balancing Alignment and Uniformity (BAU), is proposed to maintain a balance between alignment and uniformity in the latent space, along with a domain-specific uniformity loss for domain-invariant representation. The empricial results show the effectiveness of the proposed methods on various benchmarks.

**Strengths:**

- The motivation of the work is clear and well described. It is found that the data augmentations for ReID models can have negative effects on out-of-distribuiton data in despite of their positive effects on in-distribution data. This is a significant problem in domain generalizable ReID. Although this finding is not novel one, the insights provided by the analysis in terms of the alignment and uniformity concepts [1] are appreciated.
- The proposed framework is simple and effective. The proposed loss functions are designed to address the polarized effect of the data augmentation. These losses are proper application of the alignment loss and uniformity loss [1] to the domain generalizable ReID task with some modifications.
- The empricial results show that the proposed method is promising.

[1] Wang and Isola, Understanding contrastive representation learning through alignment and uniformity on the hypersphere, ICML, 2020.

**Weaknesses:**

# Major concerns
In short, I have concerns with regard to the generalization of the polarized effects by the data augmentations, and a more thorough investigation is needed in terms of the types of the backbone, loss function, and data augmentation method.
- Is the performance degradation on unseen domains due to the data augmentations generalized, regardless of model architecture and loss function? In this paper, the experiments are limited to a specific backbone and loss functions (i.e., ResNet50 trained with cross entropy and triplet losses). Would a similar polarized phenomenon appear in other backbones (e.g., ViT) or other loss functions? In this regard, while the paper mentions that the random erasing (RE) augmentation causes performance degradation on unseen domains, QAConv [2] reports performance improvement due to RE in cross-domain evaluation (Table 1 in appendix of [2]).
- Wouldn’t the polarized effect tend to be different depending on the type of data augmentation? The individual analysis on the effect of each augmentation is required. In this regard, RandAugment and Color Jitter augmentations are used by many domain generalizable ReID methods. Are they relatively safe and is there any basis for this? If so, the performance decrease on the unseen domain in Figure 1 is seen as mainly due to RE.

# Minor concerns
- I'm a little concerned about whether the comparison of the methods in Tables 3 and 4 are fair. It seems that BAU uses more abundant augmentations. Are there any significant differences in augmentations used for each method? Also, it might be necessary to check whether the polarized effect occurs when the the data augmentations used in BAU are applied to other DG methods.
- The novelty of the proposed method is somewhat limtied, since it largely depends on the alignment and uniformity loss proposed in [1]. However, the weighting strategy for alignment loss and the domain-specific uniformity loss are newly introduced, and their effectivenss are demonstrated in ablation study.
- The analysis on the increase of training cost is required. For example, the weighting strategy for alignment loss requires additional computation of Jaccard similarity of k-reciprocal nearest neighbors within a mini-batch. Is this negligible?

[1] Wang and Isola, Understanding contrastive representation learning through alignment and uniformity on the hypersphere, ICML, 2020.
[2] Liao and Shao, Interpretable and Generalizable Person Re-identification with Query-Adaptive Convolution and Temporal Lifting, ECCV, 2020.

**Questions:**

- In Equation 5, the distance between $\tilde{f_i}$ and $f_j$ is computed. How about computing distance between $\tilde{f_i}$ and $f_i$ or between $f_i$ and $f_j$?
- In Equation 8, I understood that $f_i$ can be pushed against its class prototype. However, to push against only the class prototypes that $f_i$ does not belong to would be more intuitive.

**Limitations:**

The limitations on experiments for very large domain shifts and more advanced augmentation methods are discussed.

---

> ### Author Rebuttal · Authors · 2024-08-07
>
> Dear Reviewer 4LRb,
>
> We sincerely appreciate your thorough review and are grateful for your positive remarks on the motivation and insights behind our paper. We have addressed your main concerns below.
>
> ### **Regarding the polarized effects across different backbones, losses and augmentations**
> **Backbones and loss functions**:
> We investigated the existence of the polarized effect on different types of backbones, specifically transformer (ViT-B/16) and lightweight (MobileNetV2) networks.
> We trained these models using CE+Triplet loss, with and without Random Erasing (RE) and RandAugment (RA), and the results are shown in Fig. A (a).
> Additionally, in Fig.A (b), we explored different types of loss functions, specifically ArcFace[a] and PCL[b], which are widely used for re-ID and retrieval models, on ResNet-50 with and without RE+RA.
> The results demonstrate that the polarized effect is consistently observed across all tested backbones and loss functions, suggesting that the polarized effect is not limited to the specific architecture and loss function shown in Fig. 1 of the main paper.
>
> **Augmentation types**:
> We investigated the existence of the polarized effect of individual augmentation method used in BAU: RE, RA, and CJ.
> We utilized ResNet-50 backbone with CE+Triplet loss, whose results are shown in Fig. A (c).
> We can see that the performance decrease on the unseen domain is mainly due to the polarized effects observed in RE and RA, while CJ dit not show such behavior, aligning with previous findings in the field.
> While RE and RA, which introduce significant distortions (e.g., pixel drops) to images, CJ provides simple color distortions, enhancing model robustness to variations in lighting and color conditions across unseen environments.
> We also observed that increasing the augmentation probability of individual RE and RA degrades performance and uniformity (Fig. B), confirming the polarized effect caused by these augmentations in DG ReID.
>
> In summary, we generally observe the polarized effect across different backbones, loss functions, and augmentation types, and we believe this is a more general phenomenon in DG re-ID.
> Furthermore, the proposed BAU consistently improves various baselines with different backbones and losses (Tab. B) and various augmentation types (Tab. 7 in main paper).
> Based on extensive experiments and analysis, we believe that the polarized effect in our study is a general phenomenon and that we have introduced a simple but general BAU method that can have a significant impact on this field.
>
> Regarding the discrepancy with the results reported in QAConv, it is important to note that QAConv is a DG re-ID method based on local feature matching, which may lead to a different impact of RE compared to the conventional global feature approach used in our experiments. For the feature matching task, RE can improve the model's ability to learn various local correspondences while simulating diverse occlusions [44].
>
> ### **Comparisons regarding augmentations used**
> While BAU employs a combination of augmentations (CJ, RA, RE), we would like to clarify that recent state-of-the-art methods, such as META [77] and ACL [81], utilize similar augmentations (e.g., AutoAugment and CJ).
>
> To further investigate the impact of BAU's augmentations on other DG methods, we applied BAU's augmentations to ACL based on the official implementation provided by the authors (https://github.com/peterzpy/ACL-DGReID) in Table C.
> The results show that the performance of ACL decreases when it utilizes the augmentations used in BAU, which implies that the polarized effect can also occur to other methods.
> Furthermore, BAU shows consistent performance when it utilizes the augmentations used in ACL.
> Additionally, as demonstrated in Table 7 of the main paper, BAU consistently shows superior performance across various augmentation configurations, including commonly used data augmentations (CJ+RE).
>
> In summary, BAU effectively mitigates the polarized effect of data augmentations while demonstrating robust performance across various augmentation configurations.
>
> ### **Novelty of BAU**
>
> While our method builds upon the concepts of alignment and uniformity introduced in [72], we would like to clarify that we extend these ideas from the self-supervised learning to DG re-ID task.
>
> As gratefully mentioned by the reviewer, we propose additional components tailored to the DG re-ID task, such as the novel weighting strategy and the domain-specific uniformity loss.
> Extensive ablation studies and analyses in the main paper (Tables 5-6, Figures 4-5) demonstrate the effectiveness of these components.
> Furthermore, we provide additional analysis on the weighting strategy in Figure C of the rebuttal PDF, which verifies its effectiveness across various augmentation probabilities.
>
> In summary, our work provides new insights into the polarized effect of data augmentations in DG re-ID and proposes a simple yet effective solution to mitigate this issue.
> We believe our findings will have a positive impact on subsequent research in this field, particularly when developing DG re-ID methods based on augmentation techniques, which is a straightforward solution for improving generalization.
>
> ### **Computation cost**
> The weighting strategy computes Jaccard similarity of k-reciprocal NN within a mini-batch, with time complexity $O(N^2 \log(N))$ for batch size $N$.
> Here we provide computation time comparison between baseline and BAU with the same batch size.
>
> |Method| Time(s)/Iter|
> |-|-|
> |Baseline| 0.312|
> |BAU| 0.412|
> |weight $w$ | 0.086|
>
> Considering the effectiveness of the weighting strategy (Figure C), we believe that this slight additional cost is reasonable.
>
> We further address your questions on the alignment loss and the domain-specific loss in the following comments.
>
> ---
>
> [a] ArcFace: Additive Angular Margin Loss for Deep Face Recognition, CVPR 2019
>
> [b] Prototypical Contrastive Learning of Unsupervised Representations, ICLR 2021

---

> ### Author Response · Authors · 2024-08-07
>
> Here, we provide discussions with additional experimental results according to comments in the **Questions**.
>
> ### **Regarding different strategies of alignment loss**
>
> Based on the suggestion, we investigate the impact of various alignment strategies in Table A. Specifically, we compare the performance of minimizing the feature distance between augmented and original images of the same instance ($\| \tilde{f}_i - f_i \|$), between original images of positive pairs ($\| f_i - f_j \|$), and between augmented and original images of positive pairs ($\| \tilde{f}_i - f_j \|$, Ours). The results show that our strategy, which learns invariance between original and augmented images of positive pairs, achieves the best performance.
> This result demonstrates that BAU effectively learns invariance between the original image and the augmented image.
>
> ### **Regarding domain-specific uniformity loss with corresponding class prototype**
>
> This point brought up by the reviewer is indeed correct, and we appreciate the reviewer for bringing this to our attention.
>
> In our actual BAU implementation, $f_i$ and $\tilde{f}_i$ are indeed pushed against only the class prototypes within the same domain that do not belong to its own class, aligning with your intuition.
> This approach stably encourages separation between different identities while maintaining cohesion within the same identity class.
>
> Thus, the explaination of line 214 should be:
>
> > where $\mathcal{N}(\mathbf{f})$ is the index set of *nearest prototypes of $\mathbf{f}$ that are from the same source domain and different class prototypes*, ...
>
> This approach is more effective as it encourages separation between different identities while maintaining the cohesion within the same identity class.
>
> We apologize for the lack of clarity in our original description and will ensure this is accurately reflected in our revision.
>
> ---
>
> We hope our responses have addressed your concerns. We welcome any further questions or discussions about our work.
>
> Best regards,
>
> Authors of submission 16640

---

> ### Author Response · Authors · 2024-08-13
> **A gentle reminder for reviewer-author discussion**
>
> Dear reviewer 4LRb,
>
> As the reviewer-author discussion period is coming to a close, we kindly ask if there are any remaining concerns or points about our submission that we haven't sufficiently addressed.
> We're ready to provide additional clarifications or information if needed.
>
> Once again, we appreciate your valuable efforts and feedback to strengthen our work.
>
> Best regards,
>
> Authors of submission 16440

---

> > ### Comment · Reviewer_4LRb · 2024-08-13
> >
> > Thanks for your responses. The author has answered my major concerns, and I will change the score to weak accept.

---

> > > ### Author Response · Authors · 2024-08-13
> > >
> > > Dear Reviewer 4LRb,
> > >
> > > Thank you for your thoughtful consideration and for raising the score.
> > > We are pleased that our responses have addressed your concerns.
> > >
> > > We will include the additional experimental results based on your valuable feedback (e.g., the polarized effects across different backbones, losses, and augmentations) in our revision.
> > >
> > > If you have any remaining questions or feedback, we would be glad to provide additional clarifications or results.
> > > Please don't hesitate to let us know.
> > >
> > > We sincerely appreciate your valuable feedback and your time in improving our work.
> > >
> > > Best regards,
> > >
> > > Authors of submission 16440

---

### Official Review · Reviewer_fzqs · 2024-07-12

**Soundness:** 2
**Presentation:** 2
**Contribution:** 2
**Rating:** 4
**Confidence:** 5

**Summary:**

This paper investigates the polar effects of data augmentation in the domain of generalizable person re-identification. To address the problem of augmented data degrading out-of-distribution performance, this paper proposes a Balanced Alignment and Uniformity (BAU) framework, which normalizes the representation space by maintaining a balance between alignment and uniformity.

**Strengths:**

This paper shows sufficient ablation experiments and adequate visualization results are displayed.

The study on the effect of data augmentation on generalization performance is of interest.

**Weaknesses:**

Alignment and uniformity in the author's paper are not well explained and are concepts that have been applied from other articles, this paper needs to explain the meaning of these two terms in the context of the ReID task scenario as well as the need to elaborate on the differences and connections between this paper and the source of the ideas.

The trends shown in Figure 1c of this paper are consistent across the three different domains as the proportion of data augmentation increases, but what is shown in Figure 1a is the probability of being able to both increase and decrease. The results presented by the authors are not sufficient to support their conclusions.

Is there any relevant experimental or theoretical support for this paper's claim that current generalization methods affect training stability?

This paper lacks a separate analysis of the impact of different kinds of data augmentation, and according to my previous research, adding random erasing directly reduces the generalization performance of the model.

This paper lacks experimental comparisons with more recent methods and only compares methods from 2022 and earlier.

**Questions:**

See the weakness.

**Limitations:**

The paper is not clearly written, the authors do not present their contributions well, some of the experimental results in the paper do not support their conclusions, and there is a lack of comparison with more recent methods.

---

> ### Author Rebuttal · Authors · 2024-08-07
>
> Dear Reviewer fzqs,
>
> Thank you for constructive reviews and your time and effort in evaluating our work.
> Below, we address your concerns and questions.
>
> ### **Clarification of alignment and uniformity**
> We apologize for not sufficiently explaining these concepts in the context of ReID.
>
> For ReID, alignment aims to learn similar feature representations for positive pairs (same identity) across different conditions such as various poses, viewpoints and augmentations.
> Conversely, uniformity aims to distribute feature representations uniformly across the embedding space, spreading out features of different identities and encouraging the learning of diverse visual information.
>
> While our method builds upon these concepts in [72], we extend the original idea from self-supervised learning to the DG ReID task.
> For the first time, we analyze polarized effects caused by augmentations specifically in DG ReID and adapt the concepts of alignment and uniformity to mitigate such effects.
> Furthermore, we introduce novel components specifically tailored for this task: a novel weighting strategy and a domain-specific uniformity loss.
> Extensive ablation studies in our paper (Tables 5-6, Figures 4-5) thoroughly demonstrate their effectiveness.
>
> We appreciate your feedback and will include this clarification in our revision.
>
> ### **Clarification of Figure 1**
>
> We thank the reviewer for emphasizing this crucial aspect.
>
> Figure 1a shows the polarized effect of augmentation on performance in both in-distribution (ID: M→M) and out-of-distribution (OOD: MS+C3+CS→M) scenarios with varying augmentation probabilities.
> Figure 1c highlights the uniformity of feature spaces for three domains in the OOD scenario (MS+C3+CS→M): "source (MS+C3+CS) train", "source (MS+C3+CS) test", and "target (M) test". Higher augmentation probabilities lead to less uniform feature spaces with sparse representations.
>
> It is important to note that for ID, the uniformity value (Fig. 1c) is not strictly proportional to the performance (Fig. 1a).
> Rather, our key motivation is that uniformity is crucial for performance in OOD.
> As illustrated in Figure 2 of the main paper, learning invariance with increased augmentations can result in sparse (less uniform) representation spaces that rely on dominant visual information unique to the ID data.
> While these sparse representations might generalize well to ID scenarios and thus improve performance, learning diverse, non-dominant visual information (i.e., achieving more uniformity) is vital for OOD performance in DG ReID tasks, where models must handle unseen classes from unseen domains.
>
> This motivation is further supported by our analysis of OOD performance in relation to alignment and uniformity (Figure 4), which confirms that achieving greater uniformity is essential for OOD generalization.
>
> We appreciate the reviewer's insight and will enhance the clarity of our presentation and the motivation behind our method in the revised version.
>
> ### **Regarding the training instability**
> We appreciate the valuable comment.
> In our main paper, we discussed training instability related to domain-adversarial training and meta-learning.
> The widely-used meta-learning [15] in DG methods is known to face training instability by high-order gradients [a, b].
> Similarly, several studies have highlighted stability issues in domain-adversarial training  [c, d].
> In the context of DG ReID, previous works [62, 85] have raised these issues or proposed solutions to these challenges.
> We will clarify these points and include proper references in our revised version.
>
> ### **Polarized effects across augmentation types**
> In Figure A.(c), we investigate the polarized effect of individual augmentations: Random Erasing (RE), RandAugment (RA), and Color Jitter (CJ).
> The results show that (1) RE and RA exhibit the polarized effect (improved ID, degraded OOD performance), (2) CJ does not show this polarized effect and generally improves OOD performance, which aligns with previous findings in the field.
> This is because while RE and RA introduce significant distortions (e.g., pixel drops) to images, CJ provides simple color distortions, enhancing model robustness to variations in lighting and color conditions across unseen environments.
> Fig. B confirms that higher probabilities of RE and RA lead to a more pronounced polarized effect, consistent with Fig. 1 in the main paper.
> In contrast to most existing DG ReID that primarily utilizes augmentations without polarized effects, our BAU successfully exploits the diversity introduced by augmentations regardless of polarized effects.
>
> Additionally, Fig. A(a) and (b) show that the polarized effect persists across different backbones and loss functions, suggesting it is a general phenomenon in DG ReID.
> ### **Comparison with more recent methods**
> Thank you for your valuable comments.
>
> We acknowledge advancements such as [14] and [54] in DG ReID. For instance, [14] uses large-scale unannotated videos, and [54] introduces the Part-Aware Transformer. However, direct comparison is challenging due to differences in experimental protocols. Despite this, we believe proposed BAU is complementary and could be integrated with these advanced methods. Table B shows BAU's effectiveness with unsupervised methods using pseudo-labels and ViT backbones.
>
> We have identified recent work [e] introducing a style mixing module within a ViT backbone using Protocol-1 and will include a comparison in our revised manuscript.
>
> We appreciate your feedback and are open to include comparisons with other recent methods that align with our protocols.
>
> Best regards,
>
> Authors of submission 16640
>
> ---
> [a] How to train your MAML, ICLR 2019
>
> [b] On First-Order Meta-Learning Algorithms, arXiv:1803.02999
>
> [c] Free Lunch for Domain Adversarial Training: Environment Label Smoothing, ICLR 2023
>
> [d] A Closer Look at Smoothness in Domain Adversarial Training, ICML 2022
>
> [e] Style-Controllable Generalized Person Re-identification, MM 2023

---

> > ### Comment · Reviewer_fzqs · 2024-08-11
> >
> > Thanks to the authors for answering some of my concerns, I will change the score to Borderline Reject. the main reason for this is that the actual innovations in this paper are limited and the methods compared are not new enough. In addition, as the authors state, it is the meta-learning and adversarial learning based methods that suffer from the problem of training stability, and the authors need to add these constraints to their descriptions.

---

> ### Author Response · Authors · 2024-08-12
>
> Dear Reviewer fzqs,
>
> We sincerely appreciate your reconsideration and raising the score. We will clarify the training instability associated with meta-learning and domain-adversarial learning and provide proper references in our revision. We would like to address the remaining concerns and highlight the significance of our work.
>
> ---
>
> ### **Comparison with recent methods**
> We appreciate your valuable feedback about comparison with the most recent methods. In our revision, we will include comparisons with state-of-the-art approaches published in 2023, such as ISR (ICCV 2023) [14] and StyCon (MM 2023) [a]. Please note that the results of ISR are from ResNet-50 trained on large-scale unannotated videos (47.8M person images from 74K video clips). Meanwhile, StyCon used the same training dataset as BAU (Market-1501, CUHK02, CUHK03 and CUHK-SYSU).  A comparison with more recent methods (mAP/Rank-1) is shown below:
>
> | Method            | PRID           | GRID           | VIPeR          | iLIDs          | Average        |
> |-------------------|----------------|----------------|----------------|----------------|----------------|
> | ISR (ICCV 2023)    | 70.8 / 59.7    | 65.2 / 55.8    | 66.6 / 58.0    | **91.7 / 87.6**| 73.6 / 65.3    |
> | StyCon (MM 2023)   | **78.1 / 69.7**| 62.1 / 53.4    | 71.2 / 62.8    | 84.8 / 78.0    | 74.1 / 66.0    |
> | **BAU (Ours)**     | 77.2 / 68.4    | **68.1 / 59.8**| **74.6 / 66.1**| 88.7 / 83.7    | **77.2 / 69.5**|
>
> We further compare ISR with the proposed BAU on large-scale datasets.
> We report BAU results trained under Protocol-2 (P-2) and Protocol-3 (P-3).
>
> | Method              | CUHK03          | MSMT17          | Market-1501      | Average         |
> |---------------------|-----------------|-----------------|------------------|-----------------|
> | ISR (ICCV 2023)     | 26.1 / 27.4     | 21.2 / 45.7     | 65.1 / 85.1      | 37.5 / 52.7     |
> | **BAU (P-2, Ours)** | **42.8 / 43.9** | **24.3 / 50.9** | **77.1 / 90.4**  | **48.1 / 61.7** |
> | **BAU (P-3, Ours)** | **50.6 / 51.8** | **26.8 / 54.3** | **79.5 / 91.1**  | **52.3 / 65.7** |
>
> The results show that our method remains superior to these recent approaches.
>
> In summary, our current comparison aims to be comprehensive, including recent state-of-the-art methods.
> If you have *any suggestions of other recent methods that should be compared to our work*, we would sincerely appreciate your recommendations.
>
> ---
>
> ### **Regarding innovation of BAU**
>
> We appreciate the valuable feedback in clarifying the innovations of the proposed method.
> We would like to clarify several key innovations of our work:
> - First to analyze and address the critical issue of polarized effects by data augmentations in DG ReID, which has been overlooked in previous studies.
> - Novel weighting strategy for alignment loss to assess the reliability of augmented samples.
> - Domain-specific uniformity loss for enhancing domain-invariant feature learning.
> - Enabling effective use of strong augmentations, previously challenging in DG ReID.
> - Achieving state-of-the-art performance without complex training procedures.
>
> While building on existing concepts, BAU's unique application and novel components significantly advance the field of DG ReID.
> Furthermore, as shown in Table B in 1-page PDF, we demonstrate that the proposed method is applicable to various existing methods.
> We believe this contribution will substantially impact future research in the field.
>
> We would be grateful if you could give further consideration to these points.
> We appreciate your thorough review and remain committed to improving our manuscript based on your valuable feedback.
>
> Best regards,
>
> Authors of submission 16440
>
> ---
> [a] Style-Controllable Generalized Person Re-identification, MM 2023

---

> ### Author Response · Authors · 2024-08-14
> **A gentle reminder for reviewer-author discussion**
>
> Dear Reviewer fzqs,
>
> We are deeply grateful for your time and valuable feedback throughout this review process. Your insightful comments have significantly contributed to improving our work. They have led to important clarifications and comprehensive analysis & comparisons, which greatly strengthen our work.
>
> As the discussion period is drawing to a close, we would like to kindly follow up on our previous response. We sincerely invite you to share any remaining concerns, particularly regarding the novelty of our work and the comparisons with recent methods, for your further consideration. We appreciate your feedback and would like to discuss these points further.
>
> ---
>
> In addition to our previous response, we would like to further clarify key contributions of our work, which we believe represent significant innovations in the field of DG ReID:
>
> 1. Insightful investigation of polarized effects in DG re-ID, a phenomenon previously overlooked.
> 2. A simple yet effective BAU framework that mitigates these effects without complex procedures.
> 3. Novel components tailored for this task: a weighting strategy for alignment loss and a domain-specific uniformity loss.
>
> During the discussion period, we further confirmed and strengthened our contributions through additional experiments:
>
> - We verified the commonality of the polarized effect across different backbones, loss functions, and augmentation types (Figures A and B).
> - We demonstrated the broad applicability of BAU on various baselines, including unsupervised approaches, different backbones, and loss functions (Table B).
> - We provided an in-depth analysis of our novel weighting strategy, showing its effectiveness across varying augmentation probabilities (Figure C).
>
> These additional results further validate our initial insights and the effectiveness of our proposed method, and we will incorporate this in our revision.
>
> ---
>
> Regarding the comparison with recent methods, we have conducted additional comparisons with more recent methods, ISR (ICCV 2023) and StyCon (MM 2023). We are open to further comparisons if there are other specific methods we should consider, and will incorporate this feedback in our revision.
>
> ---
>
> If there are any aspects of our work that you think could benefit from further elaboration or clarification, we would be happy to provide additional information promptly.
>
> Thank you again for your valuable contribution to our research. We look forward to any additional feedback you might have that would be valuable to our work.
>
> Best regards,
>
> Authors of submission 16440

---

### Official Review · Reviewer_Ywj9 · 2024-07-12

**Soundness:** 3
**Presentation:** 3
**Contribution:** 3
**Rating:** 5
**Confidence:** 5

**Summary:**

Although data augmentation can improve in-distribution performance, it may lead to a sparse representation space, thereby reducing out-of-distribution performance.
To address this issue, the authors proposed a simple yet effective framework, Balancing Alignment and Uniformity (BAU), which effectively regularizes the representation space by maintaining balance between alignment and uniformity.
BAU achieves state-of-the-art performance on various benchmarks and protocols, and extensive ablation studies validated the effectiveness of each component in BAU.

**Strengths:**

1.The paper demonstrates a well-structured presentation, with a clear outline that effectively communicates the core idea.

2.The paper provides sufficient experimental evidence to support the effectiveness of the proposed method.

**Weaknesses:**

1.As stated in [1], alignment and uniformity are two key properties related to the contrastive loss.
Can the polarized effects of data augmentation in DG re-ID be resolved by applying only the contrastive loss?

2.For Alignment Loss, how much does the sample that is corrupted during augmentation affect performance? This seems to be something that can be ignored.

3.The pipeline in this paper is very similar to Weak-Strong Augmentation[2], and the author needs to discuss the difference between them.

[1]Wang, Tongzhou, and Phillip Isola. "Understanding contrastive representation learning through alignment and uniformity on the hypersphere." International conference on machine learning. PMLR, 2020.

[2]Li, Yu-Jhe, et al. "Cross-domain adaptive teacher for object detection." Proceedings of the IEEE/CVF Conference on Computer Vision and Pattern Recognition. 2022.

**Questions:**

Same to the Weaknesses.

**Limitations:**

The authors have addressed the limitations and potential negative societal impact of work.

---

> ### Author Rebuttal · Authors · 2024-08-07
>
> Dear Reviewer Ywj9,
>
> Thank you for your insightful comments and positive remarks on our paper's structure and experimental evidence.
> We appreciate your feedback and have addressed your main concerns below.
>
> ### **Regarding whether polarized effects can be resolved by contrastive loss**
>
> To address whether contrastive loss alone could resolve the polarized effects of data augmentation in DG re-ID, we conducted additional experiments comparing BAU with Supervised Contrastive Learning (SupCon) [a].
> The results (mAP/Rank-1) on protocol-2 are shown in the table below:
> | Method              | CUHK03          | MSMT17          | Market-1501      | Average         |
> |-|-|-|-|-|
> | Baseline            | 33.5 / 33.7     | 16.8 / 35.9     | 63.4 / 83.0      | 37.9 / 50.9     |
> | + SupCon            | 37.9 / 37.6     | 18.2 / 41.5     | 71.8 / 87.0      | 42.6 / 55.4     |
> | + **BAU (Ours)**    | **42.8 / 43.9** | **24.3 / 50.9** | **77.1 / 90.4**  | **48.1 / 61.7** |
>
> While SupCon improves performance over the baseline, our BAU consistently outperforms SupCon across all settings.
> These results suggest that while contrastive loss can help mitigate some polarized effects, our approach of explicitly balancing alignment and uniformity is more effective in addressing this issue in the context of DG re-ID.
>
> It is worth noting that contrastive learning, including SupCon, has been shown to optimize for alignment and uniformity jointly and asymptotically [b]. In contrast, BAU directly and independently optimizes these properties, allowing for more fine-grained control and effective balancing.
>
> ### **Impact of corrupted samples during augmentation in alignment loss**
>
> To mitigate the impact of corrupted samples, we introduced a weighting strategy for the alignment loss (Eq. 4 and 5 in our paper).
> This strategy assigns lower weights on the alignment loss computed between original and potentially unreliable augmented samples, and higher weights on that computed between reliable pairs of original and augmented images, thus reducing the influence of potentially corrupted samples.
>
> To demonstrate the effectiveness of the alignment loss with the weighting strategy, we conducted additional analysis comparing the performance with varying augmentation probabilities, both with and without the weighting strategy.
> The results are shown in Figure C of the 1-page rebuttal PDF.
>
> The results show that our weighting strategy consistently improves performance across different augmentation probabilities, with the gap becoming more pronounced at higher probabilities where corruption is more likely.
> For instance, at an augmentation probability of 0.5, the weighting strategy improves the mAP from 78.3\% to 79.5\%, and at a probability of 1.0, the improvement is even more substantial, from 66.1\% to 76.1\%.
> Thanks to the proposed weighting strategy, BAU allows learning invariance with reliable augmented samples from informative augmentations.
>
> Furthermore, Tables 7 and 8 in the supplementary material of our main paper demonstrate that BAU consistently improves performance compared to the baseline for various augmentation configurations and probabilities, further validating the effectiveness of our approach in handling potentially corrupted samples.
>
> ### **Comparison with Weak-Strong Augmentation**
>
> We appreciate the constructive comments regarding the relation to existing techniques.
> Our approach is specifically designed to address the unique challenges of DG re-ID, with a particular focus on mitigating the polarized effects of data augmentation.
> While our BAU and the Cross-Domain Adaptive Teacher (CDAT) method [c] share a similar spirit in exploiting different levels of data augmentations to improve model generalizability, there are significant key differences:
>
> ***Task goal:***
> - CDAT (CVPR 22): Cross-domain adaptation for object detection.
> - BAU (Ours): Domain generalization for person re-ID.
>
> ***Framework:***
> - CDAT (CVPR 22): Utilizes a teacher-student framework where the teacher model with weak augmentations guides the student model with strong augmentations in a mutual learning manner.
> - BAU (Ours): Employs a single model framework where the model is trained with both original and augmented images while balancing alignment and uniformity.
>
> ***Techniques:***
> - CDAT (CVPR 22): Focuses on generating reliable pseudo labels by the teacher model and utilizes domain-adversarial loss with a domain discriminator.
> - BAU (Ours): Focuses on mitigating polarized effects of data augmentation in DG re-ID and utilizes domain-specific uniformity loss without any domain classifiers.
>
> In summary, BAU is specifically tailored for domain generalization in person re-ID based on our experimental observations and analysis of alignment and uniformity.
> Based on these insights, we further propose a novel weighting strategy for alignment loss and domain-specific uniformity loss to enhance model generalizability, clearly distinguishing our approach from CDAT's focus on cross-domain object detection adaptation.
>
> We hope these explanations and additional results clarify the distinctions and advantages of our approach.
> Thank you again for your valuable feedback, which has helped us provide a more comprehensive evaluation of our method in relation to existing techniques.
>
> Best regards,
>
> Authors of submission 16640
>
> ---
>
> [a] Supervised Contrastive Learning, NeurIPS 2020
>
> [b] Understanding Contrastive Representation Learning through Alignment and Uniformity on the Hypersphere, ICML 2020
>
> [c] Cross-Domain Adaptive Teacher for Object Detection, CVPR 2022

---

> ### Author Response · Authors · 2024-08-13
> **A gentle reminder for reviewer-author discussion**
>
> Dear reviewer Ywj9,
>
> As the reviewer-author discussion period is coming to a close, we kindly ask if there are any remaining concerns or points about our submission that we haven't sufficiently addressed.
> We're ready to provide additional clarifications or information if needed.
>
> Once again, we appreciate your valuable efforts and feedback to strengthen our work.
>
> Best regards,
>
> Authors of Submission 16440

---

> > ### Comment · Reviewer_Ywj9 · 2024-08-13
> >
> > Thanks for your responses. The author has answered some of my doubts, and I will maintain my score, combined with the comments of other reviewers.

---

> > > ### Author Response · Authors · 2024-08-13
> > >
> > > Dear Reviewer Ywj9,
> > >
> > > Thank you for your feedback and for taking the time to consider our responses.
> > > We're pleased that our responses have helped to address some of your concerns.
> > >
> > > If you have any remaining specific concerns or doubts, please let us know.
> > > We would be glad to provide additional clarifications or results for these points if needed.
> > >
> > > Once again, we appreciate your valuable time and comments, and we look forward to the opportunity to further improve our paper based on your insights.
> > >
> > > Best regards,
> > >
> > > Authors of Submission 16440

---

> ### Author Response · Authors · 2024-08-14
>
> Dear Reviewer Ywj9,
>
> We are deeply grateful for your time and efforts during the discussion period.
>
> Throughout this period, our further explanations with additional results have addressed the concerns raised by reviewers, leading to positive feedback which has been very encouraging for us.
>
> ---
> We would like to further clarify our improvements made during the rebuttal period:
>
> * Further investigation of polarized effects: We have conducted additional experiments exploring the commonality of polarized effects across different backbones, loss functions, and augmentation types (Figures A and B). These results provide further support for our findings and intuitions regarding polarized effects.
>
> * Broad applicability of BAU: We have investigated the effectiveness of BAU on various baselines, including unsupervised approaches, different backbones, and loss functions (Table B). These experiments validate the versatility and potential applicability of our method.
>
> * More analysis of the weighting strategy: We have provided an in-depth analysis of our novel weighting strategy, examining its behavior across varying augmentation probabilities (Figure C). This analysis presents additional insights into the robustness of our approach.
>
> * Clarification of our work: We have provided a more detailed explanation of how our method uniquely applies concepts of alignment and uniformity to the specific challenges of DG re-ID, distinguishing our approach from other existing studies.
>
> * Additional exploration of our design choices: We have conducted further analysis and comparisons in Tables A and C, which aim to demonstrate the effectiveness of BAU across different augmentation configurations and its performance relative to existing methods.
>
> We will incorporate these results into our revision.
>
> ---
>
> We are particularly thankful for your insightful comments, which have greatly contributed to enhancing our work. Your feedback inspired the valuable analysis of the alignment loss presented in Figure C and several important comparisons with existing works, which greatly strengthen our paper.
>
> As the discussion period is drawing to a close, if there are any additional points that you think could be clarified in our work, we would be happy to provide additional information promptly.
>
> Once again, we sincerely appreciate your valuable feedback and insights in improving our work.
>
> Best regards,
>
> Authors of submission 16440

---

### Official Review · Reviewer_L6wV · 2024-07-12

**Soundness:** 2
**Presentation:** 3
**Contribution:** 2
**Rating:** 5
**Confidence:** 3

**Summary:**

The authors investigate the polarized effects of data augmentations in DG re-ID and reveal that they can lead to sparse representation spaces, which are detrimental to generalization. To address it, they propose a novel BAU framework that mitigates the polarized effects of data augmentations by balancing alignment and uniformity in the representation space. And then they further introduce a domain-specific uniformity loss to enhance the learning of domain-invariant representations.

**Strengths:**

Since data augmentations can induce sparse representation spaces with less uniformity, the authors propose a simple yet effective framework, Balancing Alignment and Uniformity (BAU), which alleviates the polarized effects of data augmentations by maintaining a balance between alignment and uniformity. This work highlights the significant potential of balancing alignment and uniformity to enhance generalization for person re-identification.

**Weaknesses:**

The effects of the BAU framework are validated on one baseline model in Table 5, it will be more convincing if it is validated on three typical baseline models such as supervised, self-supervised and unsupervised models.

**Questions:**

1.Whether different alignment strategies will affected the effect of the proposed framework?
2.The effects of the BAU framework are validated on one baseline model in Table 5, it will be more convincing if it is validated on three typical baseline models such as supervised, self-supervised and unsupervised models.
3.How about the versatility of the BAU framework? Do these losses apply to the three typical models？

**Limitations:**

The authors have given the limitations of the proposed model, the authors mentioned that, as a straightforward approach primarily based on data augmentations for given input data, the method could face challenges under very large domain shifts. And they adopt standard image-level data augmentations, such as random flipping, erasing, and color jitter, and do not incorporate more advanced augmentation techniques, such as adversarial data augmentations or feature-level augmentations, which have also shown to be promising for domain generalization.

---

> ### Author Rebuttal · Authors · 2024-08-07
>
> Dear Reviewer L6wV,
>
> Thank you for your thorough review and constructive feedback. We greatly appreciate your time and effort in evaluating our work. Below, we address your concerns and questions.
>
> ### **Regarding the effects of different alignment strategies**
> We report the impact of various alignment strategies in Table A. Specifically, we compare the performance of minimizing the feature distance between:
> * Original images of positive pairs ($\| f_i - f_j \|$)
> * Augmented and original images of the same instance ($\| \tilde{f}_i - f_i \|$)
> * Augmented and original images of positive pairs ($\| \tilde{f}_i - f_j \|$, **Ours**)
>
> The results show that our strategy, which learns invariance between original and augmented images of positive pairs, achieves the best performance.
> This result demonstrates that BAU effectively learns invariance between the original image and the augmented image.
>
> Furthermore, as shown in Figure C in the 1-page rebuttal PDF, our weighting strategy for the alignment loss consistently improves the performance across different augmentation probabilities, and the effectiveness of the weighting strategy becomes more pronounced at higher augmentation probabilities.
> This indicates that focusing on reliable pairs between original and augmented images enhances the learning of robust features.
>
> We will include this analysis in the revised manuscript to provide a more comprehensive understanding of the alignment loss and the effectiveness of our weighting strategy.
>
> ### **Regarding the versatility of BAU and validation on different baselines**
> We agree that validating the effectiveness of BAU on various types of models would further strengthen our work.
> In the rebuttal,  we provide additional experimental results applying BAU to five different baselines:
> * Unsupervised approach (Cluster Contrast [a])
> * Lightweight backbone (MobileNetV2 [b])
> * Transformer-based backbone (ViT-B/16 [c])
> * Different loss functions for re-ID
>   * ArcFace [d]
>   * PCL [e, f]
>
> The results in Table B (in the 1-page rebuttal PDF) show consistent performance improvements across all these diverse baselines, demonstrating the versatility and general applicability of BAU.
> Notably, BAU achieves these improvements as a simple regularization method, without complex training procedures or additional trainable parameters.
> This highlights its efficiency and potential for easy integration across various baseline models in DG re-ID tasks.
>
> Besides, we would like to clarify that BAU is designed for supervised domain generalization settings, as it requires labels to compute the alignment loss and domain-specific loss. Therefore, directly applying BAU to self-supervised approaches is non-trivial. Nonetheless, to demonstrate the potential of BAU in unsupervised scenarios (Cluster Contrast), we adapt it to work with pseudo-labels obtained from clustering. The results demonstrate improved performance, indicating the potential for extending BAU to self-supervised and unsupervised settings without ground-truth labels in future work.
>
> In the revised manuscript, we will include the additional experimental results and discussions to provide a more comprehensive evaluation of versatility and applicability to various baselines. We will also clarify the focus on supervised learning settings and the potential for future extensions to self-supervised and unsupervised approaches.
>
> Once again, we thank you for your valuable feedback and hope that our responses and additional results address your concerns. We look forward to further improving our work based on your suggestions.
>
> Best regards,
>
> Authors of submission 16640
>
> ---
>
> [a] Cluster Contrast for Unsupervised Person Re-Identification, ACCV 2022
>
> [b] MobileNetV2: Inverted Residuals and Linear Bottlenecks, CVPR 2018
>
> [c] An Image is Worth 16x16 Words: Transformers for Image Recognition at Scale, ICLR 2021
>
> [d] ArcFace: Additive Angular Margin Loss for Deep Face Recognition, CVPR 2019
>
> [e] Prototypical Contrastive Learning of Unsupervised Representations, ICLR 2021
>
> [f] Self-paced Contrastive Learning with Hybrid Memory for Domain Adaptive Object Re-ID, NeurIPS 2020

---

> > ### Comment · Reviewer_L6wV · 2024-08-13
> >
> > Thanks for your responses. The author has answered some of my doubts, and I will change the score to borderline accept.

---

> > > ### Author Response · Authors · 2024-08-13
> > >
> > > Dear Reviewer L6wV,
> > >
> > > Thank you for your thoughtful consideration and for raising the score.
> > > We are glad that our responses have addressed some of your concerns.
> > >
> > > If you have any remaining questions or feedback, we would be glad to provide additional clarifications or results if needed.
> > > Please don't hesitate to let us know.
> > >
> > > We sincerely appreciate your valuable feedback to improve our work.
> > >
> > > Best regards,
> > >
> > > Authors of submission 16440

---

> ### Author Response · Authors · 2024-08-13
> **A gentle reminder for reviewer-author discussion**
>
> Dear reviewer L6wV,
>
> As the reviewer-author discussion period is coming to a close, we kindly ask if there are any remaining concerns or points about our submission that we haven't sufficiently addressed.
> We're ready to provide additional clarifications or information if needed.
>
> Once again, we appreciate your valuable efforts and feedback to strengthen our work.
>
> Best regards,
>
> Authors of submission 16440

---

### Author Rebuttal · Authors · 2024-08-06

Dear reviewers and chairs,

We sincerely appreciate all reviewers for their time and efforts for reviewing our work.
We are glad that the reviewers found our work  "clear motivation and idea"(Ywj9, 4LRb), "simple and effective"(L6wV, 4LRb), "well presented"(Ywj9, 4LRb), and "sufficient experimental evidence" (Ywj9, fzqs).
We believe our BAU has potential to complement the ongoing advancements in data augmentation techniques, offering opportunities for synergistic combinations to further improve model generalization.

In response to the reviewers' constructive feedback and insightful comments, we have included a 1-page PDF with additional experimental results.

This additional material addresses key concerns and provides further evidence to support our method:

* Figures A and B: Investigate ***Commonality of the polarized effect*** across different backbones, loss functions, and augmentation types. (fzqs, 4LRb)
* Table A and Figure C: Demostrate ***Effectiveness of the alignment loss*** with our proposed weighting strategy. (L6wV, Ywj9, 4LRb)
* Table B: Shows ***Applicability of the proposed method***  on various types of baselines. (L6wV)
* Table C: Validate ***Robustness of BAU*** with different augmentation configurations. (4LRb)

Overall, the attached 1-page PDF includes:

- **Figure A**: Additional investigation of the existence of polarized effects with different types of (a) backbones, (b) loss functions, and (c) augmentation types.

- **Figure B**: Additional analysis of polarized effect with varying augmentation probabilities by individual data augmentation, (a) Random Erasing and (b) RandAugment.

- **Figure C**: An analysis of the weighting strategy for the alignment loss with varying augmentation probabilities.

- **Table A**: A comparison between different strategies of alignment loss.

- **Table B**: A validation of the proposed BAU on various baseline models consisting of an unsupervised approach, different backbones, and loss functions.

- **Table C**: A comparison between recent SoTA method ACL and the proposed BAU with different augmentation configurations.

Once again, we would like to thank all reviewers for their constructive feedback, which helped us strengthen our work.

Best regards,

Authors of submission 16440

---

### Author Response · Authors · 2024-08-14

Dear reviewers and chairs,

We sincerely thank all reviewers for their responses and efforts during the discussion period.
Your valuable comments have significantly contributed to improving our work.
We are pleased that our responses have addressed concerns and received positive feedback with increased ratings, which has been very encouraging for us.

Given the short remaining time of the discussion period, we would like to kindly ask if there are any remaining questions or concerns.
We would be glad to address them with additional clarifications or results as soon as possible.

Lastly, we would like to emphasize some final points and the key contributions of our work:

1. **Insightful investigation of polarized effects in DG re-ID**: Our work is the first to thoroughly investigate and address the critical issue of polarized effects caused by data augmentations in DG re-ID. This phenomenon has been overlooked in previous studies, and our comprehensive analysis provides new insights into the field.
***
> **Note**: We have confirmed the commonality of our analysis and insights behind the polarized effect across different backbones, loss functions, and augmentation types in Figures A and B. We will include these additional results in our paper.
***
2. **Practical and effective BAU framework**: We propose a simple yet effective approach that mitigates polarized effects without requiring complex training procedures. BAU achieves state-of-the-art performance while maintaining simplicity and efficiency. Based on the concepts of alignment and uniformity, we further propose *novel components specifically tailored for this task*: a novel weighting strategy and a domain-specific uniformity loss.
***
> **Note**: In Table B, we have validated the effectiveness and broad applicability of BAU on several different baselines: unsupervised approach, other backbones, and loss functions. We will include these additional results in our paper.
***
3. **Extensive experimental results and analysis**: We have provided comprehensive experimental results including quantitative and qualitative analysis:
    * Tables 3-4 demonstrate the superior performance of BAU against other DG re-ID methods.
    * Tables 5-6 and Table A provide extensive ablation studies validating the effectiveness of each component in BAU.
    * Tables 7-8 and Table C show the robustness of our method against different augmentation configurations.
    * Figures 4-5 provide quantitative and qualitative analyses validating the effectiveness of our novel components: the weighting strategy and domain-specific uniformity loss.
***
> **Note**: Figure C further quantitatively analyzes our novel weighting strategy across varying augmentation probabilities, demonstrating its crucial role in enabling the effective use of strong augmentations. We will include these additional analysis in our paper.
***

We believe comprehensive analyses and additional insights (e.g., commonality of the polarized effects and broad applicability of the proposed method) significantly strengthen our work's impact in the field of DG re-ID.

Once again, we sincerely appreciate your time and valuable feedback in improving our work.

Best regards,

Authors of submission 16440

---

### Decision · Program_Chairs · 2024-09-25

**Decision:**

Accept (poster)

**Comment:**

This paper investigates the polarized effects of data augmentations in DG re-ID and reveal that they can lead to sparse representation spaces, which are detrimental to generalization. To address it, they propose a framework BAU that mitigates the polarized effects of data augmentations by balancing alignment and uniformity in the representation space, where a domain-specific uniformity loss is introduced to enhance the learning of domain-invariant representations.

Reviewers have appreciation for writing/presentation, sufficient experiments, clear motivation, simple and effective framework, while raising concerns on relation with contrastive loss (Reviewer Ywj9), validation on more baselines (Reviewer L6wV), limited innovations (Reviewer fzqs, 4LRb: alignment and uniform concepts were already introduced in previous work), polarized phenomenon on augmentation types (Reviewer 4LRb).

In considering the insight on the polarized effects caused by data augmentations in DG re-ID and the simple yet effective framework design,  AC arrives at an acceptance recommendation. AC highly encourages the authors to incorporate the suggestions of reviewers, add the analysis on different augmentation types, and clarify the contribution of this work in the revision.